# Operando and three-dimensional visualization of anion depletion and lithium growth by stimulated Raman scattering microscopy

Qian Cheng [1], Lu Wei[2], Zhe Liu[3], Nan Ni[1], Zhe Sang[1], Bin Zhu[1], Weiheng Xu[1], Meijie Chen[1], Yupeng Miao[2], Long-Qing Chen[3], Wei Min [2] & Yuan Yang [1]

Visualization of ion transport in electrolytes provides fundamental understandings of electrolyte dynamics and electrolyte-electrode interactions. However, this is challenging because existing techniques are hard to capture low ionic concentrations and fast electrolyte dynamics. Here we show that stimulated Raman scattering microscopy offers required resolutions to address a long-lasting question: how does the lithium-ion concentration correlate to uneven lithium deposition? In this study, anions are used to represent lithium ions since their concentrations should not deviate for more than 0.1 mM, even near nanoelectrodes. A three-stage lithium deposition process is uncovered, corresponding to no depletion, partial depletion, and full depletion of lithium ions. Further analysis reveals a feedback mechanism between the lithium dendrite growth and heterogeneity of local ionic concentration, which can be suppressed by artificial solid electrolyte interphase. This study shows that stimulated Raman scattering microscopy is a powerful tool for the materials and energy field.

[1] Department of Applied Physics and Applied Mathematics, Program of Materials Science and Engineering, Columbia University, New York, NY 10027, USA. [2] Department of Chemistry, Columbia University, New York, NY 10027, USA. [3] Department of Materials Science and Engineering, The Pennsylvania State University, University Park, State College, PA 16802, USA. These authors contributed equally: Qian Cheng, Lu Wei. Correspondence and requests for materials should be addressed to W.M. (email: wm2256@columbia.edu) or to Y.Y. (email: yy2664@columbia.edu)

on transport in electrolytes plays a crucial role in various applications, such as batteries[1–4], fuel cells[5,6], electrodeposition[7,8], and desalination[9,10]. For instance, inhomogeneous ionic flux and ion depletion near electrodes compromise the power density, operational life and safety of batteries[11–16]. One of the most crucial safety concerns is the interplay between dendrite growth and Li$^+$ ion depletion in the vicinity of a lithium (Li) metal anode. Li metal electrodes are promising for next-generation energy storage[17–20] because they have 10-times more theoretical specific capacity than commercial graphite and very negative potential (−3.04 V vs. standard hydrogen electrode)[21–23]. However, uncontrollable reduction of Li$^+$ ions usually stimulates dendrite growth, which lowers the Coulombic efficiency and causes severe safety issues, such as explosions, preventing commercialization of these systems[1,21,24,25]. Although various strategies have been applied to stabilize lithium electrodeposition[26–28], the dendrite growth mechanism, which involves ion transport in electrolytes, electrode reactions, and solid electrolyte interphase (SEI), is fairly complex and not fully understood[29,30]. A fundamental question that remains is how ion distribution and depletion affect Li deposition and morphology. Recently, Li$^+$ depletion was proposed to induce fast growth of dendritic Li filaments in zero-dimensional (0D) Li electrodes by optical imaging[14]. This conclusion was partially supported by magnetic resonance imaging (MRI) line scans with limited resolution (~0.1 mm)[12]. However, this finding has not been validated by ion concentration profile mapping of 0D, not to mention two-dimensional (2D) electrodes, which is a more important and realistic model to understand uneven deposition on Li metal.

Indeed, imaging ion transport in a liquid electrolyte is highly challenging. Electrolytes possess a much lower ionic concentration (0.01–2 M) and significantly higher diffusion coefficient (~$10^{-6}$ cm$^2$ s$^{-1}$) than a solid phase (10–50 M, <$10^{-9}$ cm$^2$ s$^{-1}$). Therefore, chemical-specific imaging with a sufficient sensitivity (better than 10 mM) and fine temporal (faster than 1 s per frame) and spatial resolution (finer than 1 μm) is required to characterize 3D ion transport in electrolytes. These requirements are beyond the capabilities of existing tools such as transmission electron microscopy[31–33], synchrotron-based techniques (detection limit ~0.2–0.5 M)[34], and MRI (~ 0.1 mm and ~10 min)[12,35]. Fluorescence microscopy has a high sensitivity and spatiotemporal resolution[36,37], but few dyes, if any, can survive the highly reducing environment near Li electrodes. Additionally, the introduction of exogenous dyes complicates the interpretation of the imaging results. In contrast, Raman spectroscopy directly targets the vibrational motions of chemical bonds in molecules in a label-free manner and should be well suited to examine Li-ion batteries[13]. Raman spectroscopy can detect [Li$^+$] by Li$^+$-solvent interactions[13,38,39] or anion concentration based on "electroneutrality"[3,39,40]. However, conventional spontaneous Raman microscopy suffers from an intrinsically weak signal and has a rather poor temporal resolution (~10 min per frame), which is not sufficient to follow rapidly changing electrolyte concentrations[38,39].

Here we exploit stimulated Raman scattering (SRS) microscopy, a nonlinear Raman technique, for operando three-dimensional visualization of ion transport in a battery electrolyte. Unlike spontaneous Raman, SRS utilizes two spatially and temporally synchronized picosecond laser pulse trains[41–43]. When the energy difference between two lasers resonates with the vibrational transition of the targeted chemical bonds, the joint action of the two laser beams can accelerate the otherwise slow vibrational transition of spontaneous Raman by $10^8$ times (Fig. 1a and Supplementary Fig. 1)[43]. Therefore, SRS microscopy offers a desirable combination of high sensitivity ( < 0.5 mM), fast imaging speed (~2 μs per pixel), fine spatial resolution (300~500 nm),

label-free nature and intrinsic 3D optical sectioning[41]. The desirable imaging capabilities of SRS have been widely applied to biomedical studies with considerable impact,[44–47] but SRS has rarely been used in material and energy studies.

In this report, we show that SRS imaging can quantitatively capture the fast evolution of the ion concentration at a Li surface and how the concentration is related to the growth of Li dendrites. Our imaging results revealed the Li electrodeposition has three stages. In addition to the slow growth of mossy Li in the initial stage and rapid dendritic growth in the final stage upon full depletion of Li$^+$, a previously unknown intermediate stage corresponding to the partial depletion of Li$^+$ was observed in a spatially heterogeneous manner. In this stage, the high local Li$^+$ concentration ([Li$^+$]) on a 2D Li electrode promotes the fast local growth of Li and initiates a positive feedback loop between the Li growth rate and [Li$^+$] heterogeneity, which leads to dendrite eruption upon full depletion. Inspired by these results, the effectiveness of a solid electrolyte coating and electrolyte additives were also investigated. A Li$_3$PO$_4$ solid electrolyte coating was observed to homogenize the ionic distribution and block the positive feedback loop, leading to a more uniform deposition of Li even with Li$^+$ depletion. Such results provide new insights into battery safety and show that SRS microscopy is a powerful technique for imaging ion transport that can be extended to many systems.

## Results

**Observation of a three-stage lithium deposition process**. To simultaneously study Li$^+$-ion transport and dendrite growth, we used a Li/gel electrolyte/Li symmetric cell as a model (Fig. 1b, c). In such a cell, the gel electrolyte is composed of lithium bis (oxalato)borate (LiBOB) in tetraethylene glycol dimethyl ether (TEGDME) with 22 wt% poly (vinylidene fluoride-cohexa-fluoropropylene) (PVdF-HFP). PVdF-HFP gel was selected to minimize the convection and electro-osmotic effect in the liquid electrolyte, enabling quantitative interpretation of concentration changes.

Although [Li$^+$] can be quantitatively detected by the Raman peaks of Li$^+$-solvent interactions (Fig. 2a), these peaks are typically below 1000 cm$^{-1}$, out of the detection window of the existing setup in the authors' laboratory. Therefore, in this study, the [Li$^+$] in the electrolyte is represented by the local Raman intensity of BOB$^-$ since the difference between [Li$^+$] and [BOB$^-$] is much less than 0.1 mM on a scale of 100 nm or above, which is well below our resolution (~10 mM). Such "electroneutrality" originates from the Poisson equation:[39,40]

$$\nabla^2 \Phi = -\frac{F}{\varepsilon_r \varepsilon_0}(c_+ - c_-) = -1.40 \times 10^6 (c_+ - c_-)(\text{V} \cdot \mu\text{m}^{-2}),$$

(1)

where $\Phi$ is the electrode potential, $F$ is the Faraday constant (96,485 C mol$^{-1}$), $\varepsilon_r$ is the relative permittivity (7.8 for TEGDME),[48] $\varepsilon_0$ is the vacuum permittivity (8.85 × $10^{-12}$ F m$^{-1}$), and $c_+$ and $c_-$ are the cation and anion concentrations in mol L$^{-1}$, respectively. For example, with an extremely high electric field gradient in the electrolyte, e.g., 5 V μm$^{-2}$, the difference between [Li$^+$] and [BOB$^-$] is >5 μM, which is negligible in our study (Supplementary Note 1). Even on nanoelectrodes with a diameter of 10 nm, both the analytical results and simulations show that the concentration difference between Li$^+$ and BOB$^-$ is <0.1 mM (Fig. 2b and Supplementary Note 1). Moreover, we also use spontaneous Raman to simultaneously track [Li$^+$] by Li$^+$–solvent interactions and [BOB$^-$] during lithium deposition (Fig. 2c, d); the measured [Li$^+$] and [BOB$^-$] near lithium dendrites agree well with each other. The difference (<2 mM) is well below the

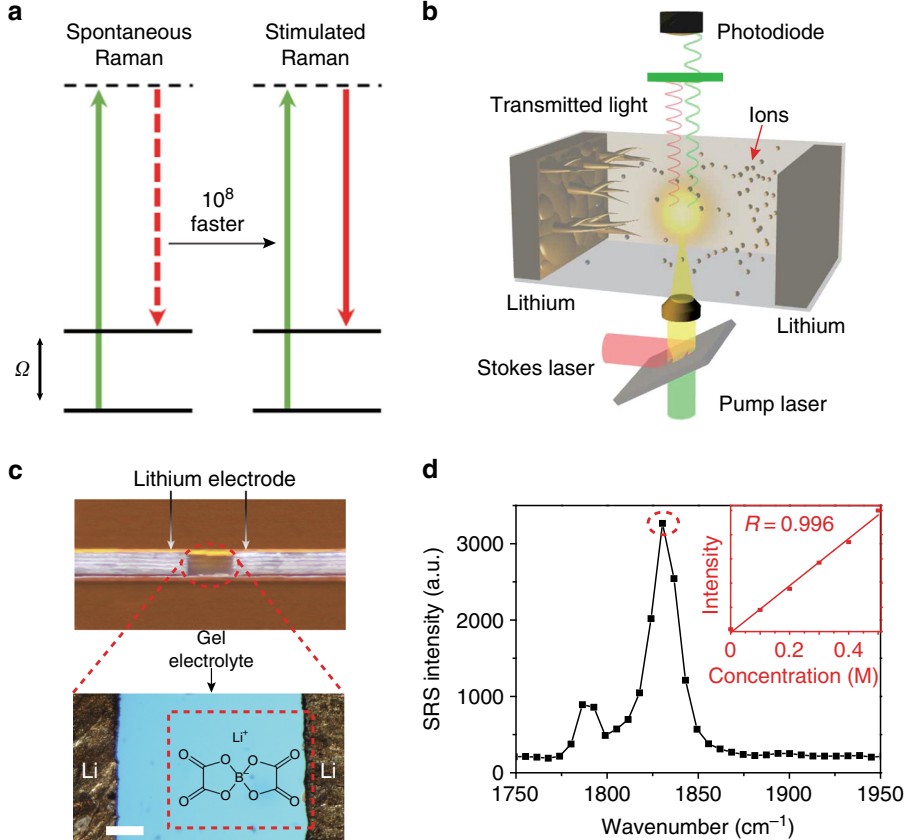

**Fig. 1** Experimental principle and design. **a** Energy diagrams of spontaneous Raman scattering and stimulated Raman scattering (SRS). In spontaneous Raman, only one laser (green, solid line) is used, and the scattered photons (red, dashed line) will have an energy loss of $\Omega$, corresponding to the vibrational energy of the targeted bond. In SRS, two different lasers (green/red, solid line) with energy gaps matching $\Omega$ are simultaneously used and yield up to $10^8$ times faster vibrational transitions. **b** A schematic illustration of a Li–Li symmetric cell under SRS imaging. The two lasers are the pump laser and Stokes laser. **c** A camera image of a Li–Li symmetric cell (top) and zoom-in microscope image (bottom). Scale bar is 100 μm. The red, dashed rectangle indicates the imaging area. The molecular structure is of the LiBOB salt used in our study. Scale bar is 100 μm. **d** The SRS spectrum of 0.5 M LiBOB in TEGDME/PVdF-HFP gel electrolyte. The inset shows the linear concentration dependence between the Raman intensity at 1830 cm$^{-1}$ (dashed circle) and the LiBOB concentration from 0 M to 0.5 M

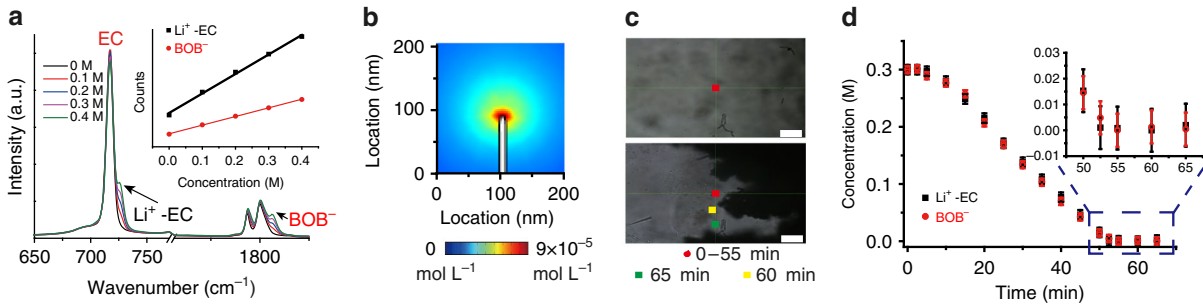

**Fig. 2** Experimental and simulation results for validating "charge neutrality". **a** The spontaneous Raman spectra of LiBOB / (TEGDME: EC v/ 7:3) electrolyte with a concentration from 0 M to 0.4 M. Inset is the plot of counts at the designated wavenumber (725 cm$^{-1}$ for Li$^+$-EC solvent solvation and 1830 cm$^{-1}$ for BOB$^-$) versus the concentration of LiBOB. **b** COMSOL finite element simulation on a 10 nm wide electrode tip during lithium reduction to show that the concentration difference between the cation and anion is less than 0.09 mM. A zoomed-out image can be found in Supplementary Fig. 2. **c** Optical images of the Li electrode at the beginning and end of lithium electrodeposition. The colored squares show positions where the Raman spectra were taken, which is near a lithium dendrite tip. Corresponding Raman spectra and voltage profile can be found in Supplementary Fig. 3. White scale bars are 50 μm. **d** The concentration changes in Li$^+$ and BOB$^-$ vs. time at the locations and times in **c**. The average difference between [Li$^+$] and [BOB$^-$] at 16 points is well below 2 mM at a noise level of 8.3 mM for Li$^+$-EC solvent solvation and 5.8 mM for BOB$^-$

instrument noise level (~ 8 mM), further confirming the validity of using [BOB$^-$] to derive the local [Li$^+$] at the given resolution (10 mM and 500 nm, Supplementary Note 2, Supplementary Fig. 3). In addition to obeying "electroneutrality", ion–ion and

ion–solvent interactions in electrolytes, such as solvent-separated ion pair (SSIP), contact ion pair (CIP) or aggregate (AGG), will not affect the study as the Raman intensity is still linearly related to the ion concentration (Supplementary Note 3, Supplementary

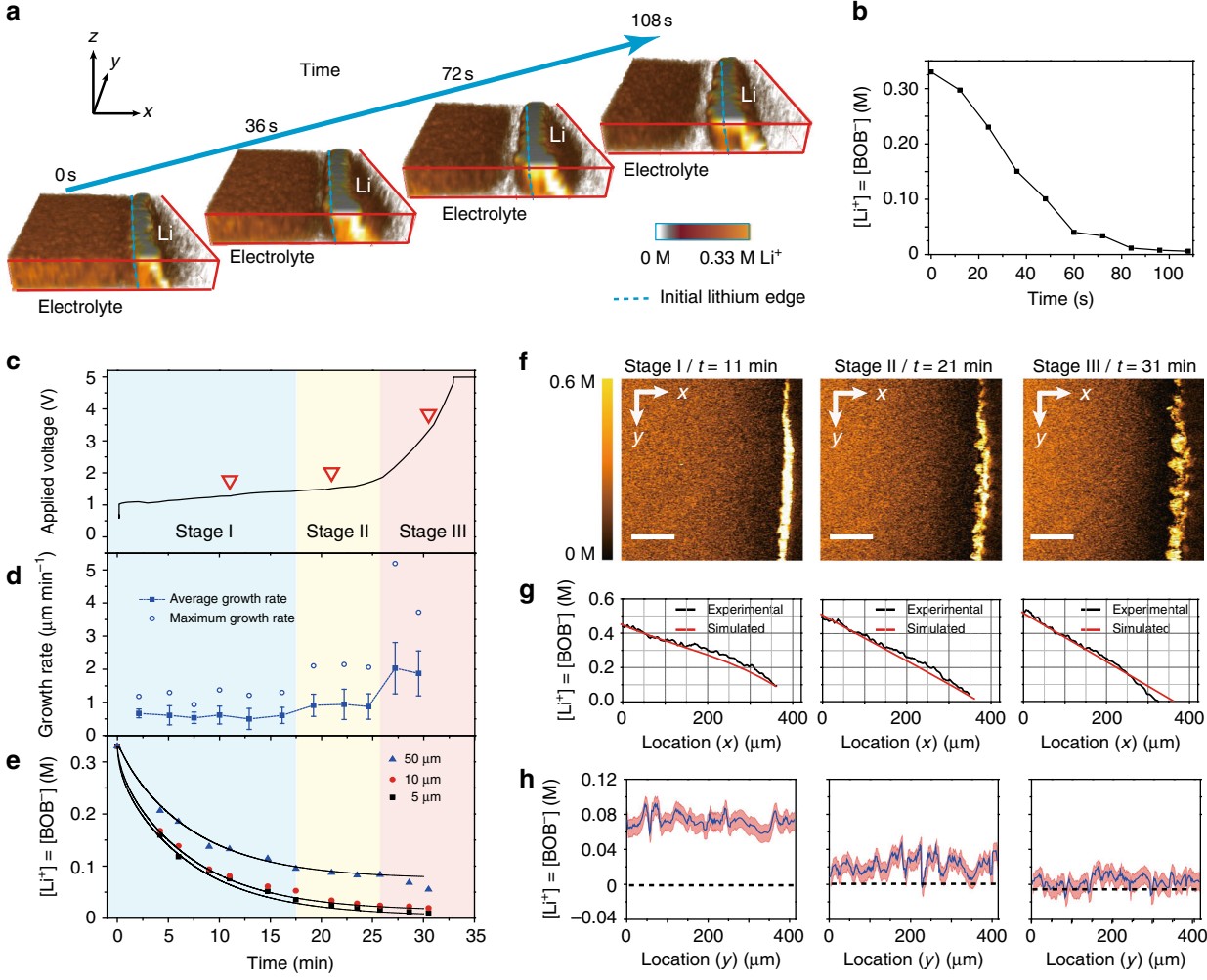

**Fig. 3** Visualization of ion transport/depletion and dendrite growth on a 2D Li electrode. **a** 3D images showing depletion of ions near the Li surface at a current density of 4.2 mA cm$^{-2}$. The expansion of the transparent area (white) near the Li electrode indicate gradual Li$^+$-ion depletion in the vicinity of the electrode. The imaging area is 150 μm ($x$) × 250 μm ($y$) × 50 μm ($z$), and the area is stretched by 66.7% along the $x$-axis to amplify the Li growth. **b** Average Li$^+$ concentration 5 μm away from the Li surface in **a**. **c** Voltage profile of a Li/Li symmetric cell at 1.3 mA cm$^{-2}$. **d** Li growth rate and standard deviation vs. time. The solid and empty squares represent the average growth rate ($v_{ave}$) and the maximum local growth rate ($v_{max}$). **e** Average [Li$^+$] vs. time at 5 μm, 10 μm and 50 μm away from the electrode surface. Black lines are the fitting curves based on the Nernst–Planck equation. **f** Representative 2D SRS images at 11 min, 21 min and 31 min. The color bar on the left represents the [Li$^+$] in the electrolyte. Scale bars are all 100 μm. **g** 1D [Li$^+$] profile along the $x$-direction of the SRS 2D images in **f** (black lines). Red lines are the simulation results based on the Nernst–Plank equation. **h** Extrapolated [Li$^+$] on the Li surface ([Li$^+$]$_{0\ \mu m}$) along the electrode contour versus the location ($y$-direction). The value is determined by a linear extrapolation of all the points between [Li$^+$]$_{10\ \mu m}$ and [Li$^+$]$_{3\ \mu m}$ (Supplementary Fig. 10). The error bar in **f** is ±10 mM. All [Li$^+$] in this study were measured based on [BOB$^-$], and their difference should be less than 0.1 mM, which is much smaller than the SRS resolution in the electrolyte

Fig. 4). For simplicity, we will use [Li$^+$] instead of [BOB$^-$] in the text as the above evidence indicates that this replacement will not distort our results, but a difference of up to 0.1 mM may exist between the two concentrations. SRS imaging can also be readily applied to other vibrational modes <1000 cm$^{-1}$ (e.g., Li$^+$-EC[13], Li$^+$-DMC[38], PF$_6^-$, and TFSI$^-$) (Supplementary Note 4).

To demonstrate fast and quantitative imaging of ion dynamics, we first verified the SRS signal dependence on the LiBOB concentration with a fast acquisition speed (32 μs per pixel) and low laser powers of $P_{pump}$ = 24 mW and $P_{stokes}$ = 50 mW. These acquisition conditions only cause a slight temperature increase of 2–5 K, which was in situ measured by a thermocouple inside a Li/Li cell. In addition, no obvious dendrite damage was observed. As shown in Fig. 1d, the measured SRS signal is proportional to the ion concentration with the minimal background noise (Supplementary Fig. 5). The detection limit is as low as 10 mM at this scanning rate and power based on a signal-to-noise ratio of 1

(Supplementary Note 5). These results indicate the high sensitivity of SRS microscopy and lay a solid foundation for our imaging and quantitative analysis below. Such high sensitivity allows a 3D volumetric image to be taken within 10 s (2 μs per pixel, 256 × 256 pixels per image, 16-time frame average and 5 $z$ depths). As shown in Fig. 3a, b (4.2 mA cm$^{-2}$ applied), the time-dependent depletion of LiBOB near a Li metal surface was nicely captured by 3D SRS imaging. In comparison, a spontaneous Raman microscope may take > 10 min to accumulate a single 35 × 35 pixel image[49]. Interestingly, the Li metal edge is nicely resolved with high intensity in our SRS images (Fig. 3a). Such non-Raman-specific cross-phase contrast might originate from the nonlinear pump-probe process at the structural edge. The position and shape match well with those from the bright-field images (Supplementary Fig. 6), indicating that this contrast can be used to outline the Li metal boundary. Hence, our technique allows simultaneous monitoring of gradual ion depletion near the

Li electrode and uneven Li deposition. We subsequently confirmed that SRS measurements of LiBOB depletion near a Li metal surface are barely affected by presence of Li dendrites (Supplementary Fig. 7).

After validating the method, we attempted to understand how [Li$^+$] near a Li metal surface interacts with Li growth. We monitored the Li negative electrode and adjacent electrolyte region by SRS (Supplementary Movie 1) with a current density of 1.3 mA cm$^{-2}$ until 5 V and then with constant voltage charging (Fig. 3c). The distance between the two Li electrodes was 450 μm, and the initial [Li$^+$] was 0.33 M because a higher concentration leads to salt precipitation on the counter electrode (Supplementary Fig. 8). As shown in Fig. 3c, once the current was applied, the voltage increased to 1.0 V due to electrolyte resistance and charge-transfer resistance. The voltage then increased slowly during the first 26 min of charging and quickly shot up to 5 V between 26 and 33 min. Consistent with previous reports[12,14], this voltage increase was accompanied by the accelerated growth of Li dendrites (Fig. 3d). Moreover, we observed that ions were gradually depleted near the Li surface (Fig. 3e, f). The observed dynamic change in the ion concentration along the x-direction is consistent with the simulations based on the Nernst–Planck equation (Fig. 3g, Supplementary Note 6). The diffusion coefficient derived from the simulation is $4.5 \times 10^{-7}$ cm$^2$ s$^{-1}$, which matched well with our experimental results ($4.9 \times 10^{-7}$ cm$^2$ s$^{-1}$) (Supplementary Note 7 and Supplementary Table 1). All [Li$^+$] in this study was measured based on [BOB$^-$], and their difference should be <0.1 mM, which is much smaller than the SRS resolution in the electrolyte (Supplementary Note 1, 2).

Since SRS can simultaneously monitor the morphology of the solid Li phase and the Li$^+$ distribution, the local Li growth rate (v) can be calculated for each point on the Li surface along with the local [Li$^+$]. In the following analysis, the [Li$^+$] values at x μm away from the surface is denoted as [Li$^+$]$_{x\,\mu m}$. The [Li$^+$] on the Li surface ([Li$^+$]$_{0\,\mu m}$, Fig. 3h) was calculated based on a linear extrapolation using all measured points between [Li$^+$]$_{3\,\mu m}$ and [Li$^+$]$_{10\,\mu m}$ (Supplementary Fig. 10) because quantitatively determining the Raman intensity at the Li surface is difficult.

We observed three distinct dynamic stages of Li growth (Fig. 3d). (I) In the first stage, i.e., slow growth (0–17 min), the average growth rate (v$_{ave}$) is ~0.6 μm min$^{-1}$, and the maximum growth rate (v$_{max}$) is ~1.2 μm min$^{-1}$. The electrode surface is nearly flat (Fig. 3f, t = 11 min), and the morphology is dominated by mossy Li. [Li$^+$]$_{0\,\mu m}$ is well above zero (Fig. 3h, t = 11 min). Hence, the concentration overpotential is low, and the corresponding voltage profile is nearly flat. (II) The faster growth stage (17–26 min) has a slightly higher v$_{ave}$ of ~0.9 μm min$^{-1}$ and a much higher v$_{max}$ of ~2.1 μm min$^{-1}$. Dendritic Li protrusions start to appear during this stage (Fig. 3f, t = 21 min), and Li$^+$ depletion partially occurs near the electrode (Fig. 3h, t = 21 min). The depletion is usually observed at the valleys on the Li surface and not the protrusions. Therefore, we define this stage as the "partial depletion" stage. The reduced active deposition area due to Li$^+$ depletion and the lower [Li$^+$] on the Li surface lead to a larger overpotential, and the cell voltage begins to increase. (III) The rapid growth of dendritic Li (after 26 min). This stage has a much higher v$_{ave}$ of 2.0 μm min$^{-1}$ and a significantly higher v$_{max}$ of 3.7–5.2 μm min$^{-1}$, which illustrate that dendritic growth dominates this stage (Fig. 3f, t = 31 min). In this stage, [Li$^+$]$_{0\,\mu m}$ is close to 0 mM on nearly all surfaces (Fig. 3h, t = 31 min), and the concentration overpotential dramatically increases and leads to the voltage increasing to 5 V.

The observed stages I and III are consistent with the previous two-stage theory[12,14] that dendrite growth will not thrive without the depletion of ions (Supplementary Note 8). However, in

addition to them, we unveiled a critical transitional stage (stage II) during which dendrite growth starts because heterogeneous depletion is much more noticeable in our 2D electrodes than the 0D electrodes used in past studies. The 2D electrode configuration is also more realistic and can better represent the deposition process in real Li electrodes.

**Positive feedback mechanism.** Realizing the importance of inhomogeneous ion distribution during Li growth, we next explored how Li deposition is affected by ion heterogeneity and depletion throughout the three stages (Supplementary Movie 2). A small current of 0.6 mA cm$^{-2}$, which is below the limiting current (0.75 mA cm$^{-2}$), was initially applied and followed by a jump to 0.9 mA cm$^{-2}$ to fully deplete Li$^+$ at the Li surface. The corresponding voltage profile is presented as Supplementary Fig. 11. Three representative cases are studied for stage (I) (35/40 min, Fig. 4a–c), (II) (65/70 min, Fig. 4d–f) and (III) (100/105 min, Fig. 4g–i). Each case includes a comparison between two adjacent time spots (t1 and t2) separated by an interval of five minutes. Using stage I as an example, the first image (Fig. 4a) shows the solid Li electrode at t1 = 35 min in turquoise with the corresponding [Li$^+$] distribution above it. The second image (Fig. 4b) shows Li at t1 in turquoise, its growth between t1 to t2 in dark gray, and the [Li$^+$] distribution at t2. The arrows in the electrolyte region in Fig. 4a, b represent the local concentration gradient. The third plot indicates the local Li growth rate (black) and [Li$^+$]$_{10\,\mu m}$ at t1 (blue) and t2 (red). Although the Raman signal is minimally affected by its position relative to the Li dendrites (Supplementary Fig. 7), a gap of 10 μm was still chosen to ensure the accuracy of the concentration. We also want to emphasize that [Li$^+$]$_{10\,\mu m}$ is a highly relevant indicator of heterogeneous Li dendrite growth because ions move at a speed of ~1–10 μm s$^{-1}$ at a given [Li$^+$] near the Li surface and current density, and the characteristic time scale for apparent morphological change during Li deposition is 1–10 s (Supplementary Note 9).

Analysis on the spatial heterogeneity of [Li$^+$] and Li growth unveils a positive feedback mechanism: the spatial heterogeneity of [Li$^+$] and flux near the Li surface promotes uneven Li growth, and the newly formed Li protrusions move into regions with a higher [Li$^+$] (stage II), which in turn amplifies [Li$^+$] and ionic flux heterogeneity and accelerates the catastrophic process (stage III). Our conclusion is supported by the various phenomena illustrated in Fig. 4. The direct evidence is as follows. In regions with fast Li growth (e.g., location (y): 112–132 μm), once depletion starts, the Li growth rate (v) speeds up and [Li$^+$]$_{10\,\mu m}$ simultaneously increases, indicating that Li dendrites approach the region with a higher [Li$^+$]; and thus, the growth is accelerated (Fig. 4j). In contrast, if [Li$^+$]$_{10\,\mu m}$ is low (location (y): 0–40 μm), Li growth is always low (Fig. 4k). The analysis of more regions is shown in Supplementary Fig. 12.

In addition to the above evidence, we also observed three additional supportive phenomena. First, a high and increasing correlation between the spatial distribution of v and [Li$^+$]$_{10\,\mu m}$ is observed. For example, the correlation coefficient (R) of v and [Li$^+$]$_{10\,\mu m}$ at t2 increases from 0.29 in stage I to 0.59 in stage II and 0.84 in stage III (Table 1 and Supplementary Fig. 13), which are all much higher than 0.22, i.e., the R value for a confidence level of 0.001. The increasing R value indicates that the above feedback mechanism strengthens the correlation between v and [Li$^+$]$_{10\,\mu m}$. The high R values in stage II and III also suggest that [Li$^+$]$_{10\,\mu m}$ is a good indicator of dendrite growth as [Li$^+$]$_{0\,\mu m}$ is nearly zero at depletion, so [Li$^+$]$_{10\,\mu m}$ is proportional to the diffusive ionic flux. Therefore, high [Li$^+$]$_{10\,\mu m}$ leads to faster local Li growth, especially in stages II and III.

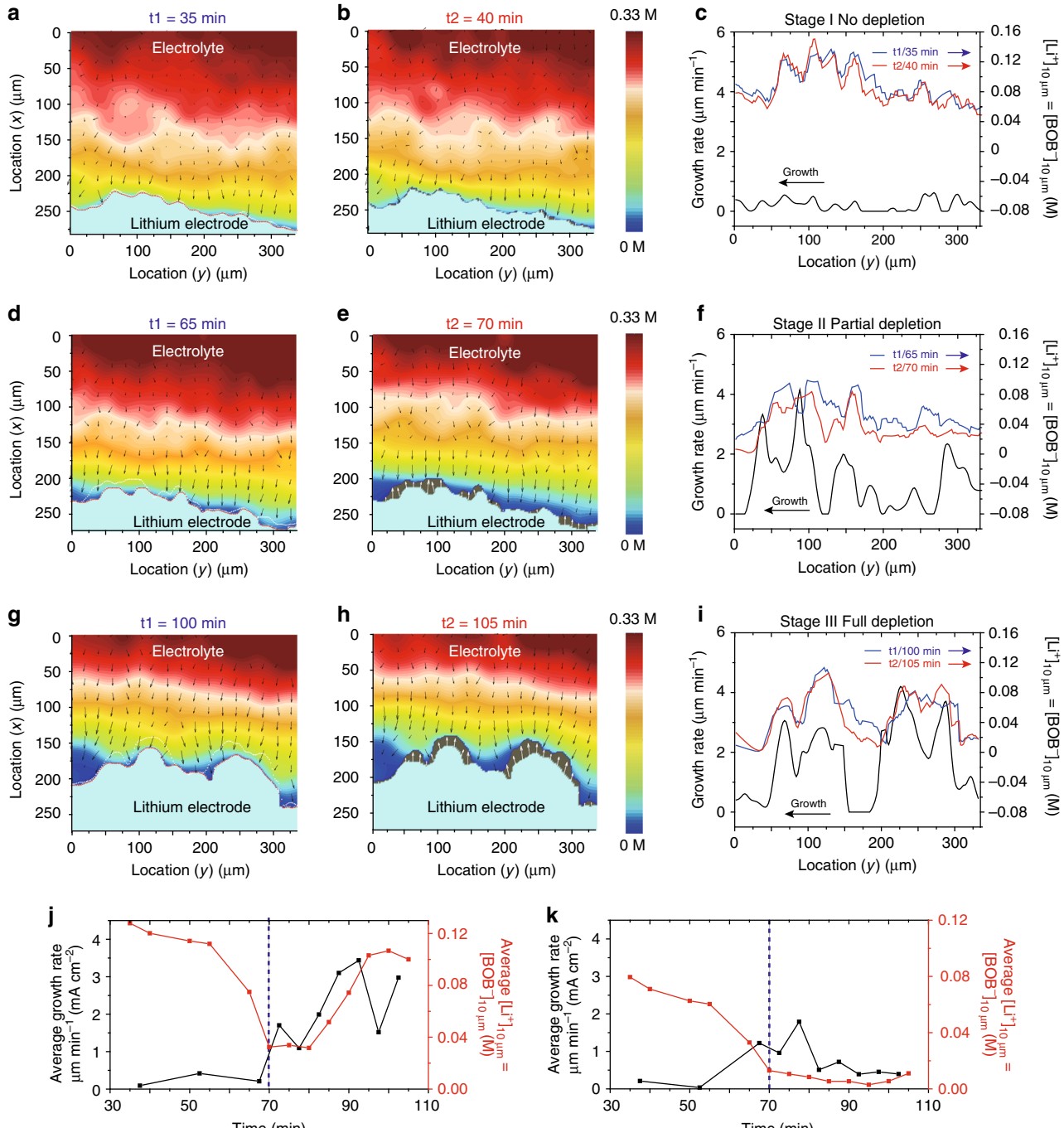

**Fig. 4** Correlation between local Li growth and local Li+ concentration. 2D overlapping images at three representative moments (t1/t2), **a–c** 35/40 min (stage I / no depletion), **d–f** 65/70 min (stage II / partial depletion), and **g–i** 100/105 min (stage III / full depletion). The first column exhibits the ionic concentration profile and solid Li electrode region at t1. The boundaries of the lithium electrode at t1 and t2 are labeled by red and white dash lines, respectively. The second column exhibits the $[Li^+]$ at t2. The gray region at the bottom shows the difference in the solid Li electrode between t1 and t2. The white arrows represent the dendrite growth direction and rate. The $[Li^+]$ in the electrolyte is represented by a jet color bar. The arrows in the electrolyte represent the concentration gradient in the electrolyte. Kernel smoothing was applied in **a–f** to reduce the noise to <10 mM (See "Methods" section and Supplementary Fig. 9). The third column (**c**, **f**, **i**) shows the relationship between the dendrite growth rate (black lines) and $[Li^+]_{10\,\mu m}$ vs. the location at t1 (blue lines) and t2 (red lines). The corresponding scattering plots are presented in Supplementary Fig. 15. **j**, **k** The relationship between the Li growth rate and $[Li^+]_{10\,\mu m}$ in two regions: fast growth region (**j**: location (y) 112–132 μm) and slow growth region. (**k**: location (y) 0–40 μm). Blue dashed line is at 70 min when depletion starts. All $[Li^+]$ values in this study are measured based on $[BOB^-]$, and their difference should be less than 0.1 mM, which is much smaller than the SRS resolution in the electrolyte

Second, the *R* value of *v* and $[Li^+]_{10\,\mu m}$ at t2 is always higher than that at t1 for all three stages (Table 1), which indicates that faster local Li growth tends to move into the region with higher $[Li^+]$, further amplifying the heterogeneity in $[Li^+]$ near the Li surface and the ionic flux. Based on such a feedback mechanism, increasing fluctuations in both *v* and $[Li^+]_{10\,\mu m}$ from stage I to III are expected, which is the third phenomenon observed (Table 1). In stage I, the standard deviation of *v* ($v_{SD}$) and $[Li^+]_{10\,\mu m}$ ($c_{SD}$) is

| Table 1 Correlation between the local Li growth rate and Li$^+$ concentration heterogeneity | | | No depletion (35/40 min) | Partial depletion (65/70 min) | Full depletion (100/105 min) |
|---|---|---|---|---|---|
| Experimental results | $v$ ($\mu$m min$^{-1}$) | | 0.20 ± 0.18 | 1.14 ± 0.96 | 1.62 ± 1.23 |
| | $\overline{[Li^+]}_{10\,\mu m}$ (mM) | t1 | 91 ± 23 | 55 ± 21 | 50 ± 30 |
| | | t2 | 86 ± 24 | 37 ± 21 | 48 ± 31 |
| | $R$ of $v$ and $[Li^+]_{10\,\mu m}$ in Fig. 4 | t1 | 0.05 | 0.22 | 0.58 |
| | | t2 | 0.29 | 0.59 | 0.84 |
| Simulation results | $v$ ($\mu$m min$^{-1}$) | | 0.25 ± 0.30 | 0.42 ± 0.45 | 0.86 ± 0.82 |
| | $\overline{[Li^+]}_{10\,\mu m}$ (mM) | t1 | / | / | / |
| | | t2 | 81 ± 17 | 57 ± 19 | 46 ± 26 |
| | $R$ of $v$ and $[Li^+]_{10\,\mu m}$ in Supplementary Fig. 13 | t1 | 0.30 | 0.48 | 0.78 |
| | | t2 | 0.45 | 0.73 | 0.86 |

0.18 $\mu$m min$^{-1}$ and 23 mM, respectively, suggesting weak correlation and feedback; then, in stage II, $v_{SD}$ increases to 0.96 $\mu$m min$^{-1}$. Although $c_{SD}$ is 21 mM in stage II due to a significantly reduced average $[Li^+]_{10\,\mu m}$, the ratio of $c_{SD}$ to $c$ increases dramatically from 25% in stage I to 38% in stage II. Finally, in stage III, $v_{SD}$ and $c_{SD}$ increase to 1.23 $\mu$m min$^{-1}$ and 30 mM, respectively, which corresponds to the eruption of dendritic growth and severe heterogeneous ionic distribution.

The positive feedback mechanism was also verified by the phase-field simulations of images in Fig. 4 (Supplementary Note 10, Supplementary Fig. 14), and the results are listed in Table 1. First, the $R$ values between $v$ and $[Li^+]_{10\,\mu m}$ increase for both t1 and t2 from stage I to stage III, indicating a strong correlation between the Li growth rate and local Li$^+$ concentration. Second, the $R$ value of $v$ and $[Li^+]_{10\,\mu m}$ at t2 is always higher than the $R$ value of $v$ and $[Li^+]_{10\,\mu m}$ at t1. Third, the growth rate continues to increase from 0.25 ± 0.30 $\mu$m min$^{-1}$ (stage I) to 0.86 ± 0.82 $\mu$m min$^{-1}$ (stage III) as the $c_{SD}$ of $[Li^+]_{10\,\mu m}$ at t2 increases from 17 mM to 26 mM. The consistency between the experimental and simulation results confirms the positive feedback mechanism of catastrophic Li dendrite growth.

**Homogenizing the lithium ion concentration to suppress dendrite growth.** Based on the above results, we hypothesized that homogenizing the ion concentration and flux on a Li surface might suppress dangerous Li dendritic growth in stage II (partial depletion) and III (full depletion), which can occur in commercial batteries (Supplementary Fig. 15). To test the hypothesis, we examined two strategies: artificial SEI and electrolyte additives. A ~100 nm-thick Li$_3$PO$_4$-based solid electrolyte-like artificial SEI was used in the first strategy[50]. The current density is 1.4 mA cm$^{-2}$ (Supplementary Movie 3). The voltage profile of the Li$_3$PO$_4$-protected Li–Li cell shows behavior (Fig. 5a) similar to that of the bare Li electrodes (Fig. 3c), and the voltage begins to increase when the Li$^+$ ions are depleted at the surface.

The Li growth rate on this electrode is steady and low without obvious dendrite formation (Fig. 5b), regardless of Li$^+$ ion depletion (Fig. 5c). The $v_{ave}$ is always ~0.2–0.9 $\mu$m min$^{-1}$ and mostly ~0.4 $\mu$m min$^{-1}$; $v_{max}$ is always <1.8 $\mu$m min$^{-1}$ (Fig. 5b). Such speed is similar to that of stage I (no depletion) for bare Li electrodes but much slower than stage III (full depletion). More importantly, $[Li^+]_{10\,\mu m}$ becomes more homogeneous over time as $c_{SD}$ decreases from 18 mM (11 min) to 10 mM (31 min) and $v_{SD}$ remains at ~0.25 $\mu$m min$^{-1}$ (Fig. 5b & Supplementary Fig. 16). A possible mechanism is that the Li$_3$PO$_4$ layer acts as a buffer for Li$^+$ transport, which homogenizes the true ionic flux that the Li metal surface sees, which is supported by the simulations (Supplementary Fig. 17). In addition, Young's modulus of the Li$_3$PO$_4$ layer can reach 10 GPa, which is strong enough to

suppress Li protrusion and lead to uniform Li deposition[50]. Therefore, the positive feedback mechanism is effectively inhibited (Fig. 5d–g). This argument is also supported by the weak correlation observed between the Li growth rate and $[Li^+]_{10\,\mu m}$ (e.g., $R = 0.21$ at stage III, Supplementary Fig. 16i).

We also investigated the effect of adding an electrolyte additive, 0.05 M tetrabutylammonium hexafluorophosphate (TBA-PF$_6$). TBA$^+$ ions are not reduced during Li deposition, and as a supporting electrolyte, these ions may help shield the local electric field, reducing ion flux heterogeneity[51–53]. In an electrolyte with multiple components, Li$^+$ depletion does not require the [BOB$^-$] to vanish at the same time, but Li$^+$ depletion should still occur at approximately 30 min based on the voltage profile (Supplementary Fig. 18). After being normalized by the current, the Li growth rate in stage I ($v_{ave} < 0.3$ $\mu$m min$^{-1}$ and $v_{max} < 0.9$ $\mu$m s$^{-1}$) is much less than that of the bare Li and Li with a Li$_3$PO$_4$ coating electrode (Supplementary Fig. 19). However, the growth rate increases dramatically to $v_{ave}$ of ~1.1 $\mu$m min$^{-1}$ and $v_{max}$ of ~2.7 $\mu$m min$^{-1}$ upon full ion depletion. Such behaviors indicate that the addition of TBA-PF$_6$ can result in more compact Li deposition in stage I, which may be a result of shielding effects, but TBA-PF$_6$ cannot efficiently suppress dendritic Li growth when the [Li$^+$] depletes in stages II and III.

## Discussion

The dynamic depletion of Li$^+$ ions and growth of Li dendrites on a Li electrode surface were imaged by the anion concentration with SRS microscopy with a resolution of 10 mM, 500 nm, and < 1 s per frame since "electroneutrality" requires the concentrations to be within 0.1 mM for the spatial scale utilized. The correlations among the local ion distribution, Li growth rate, and voltage were visualized and quantitatively studied. Our work revealed that three dynamic stages exist in the Li deposition process: slow deposition of mossy Li when surface ions are not depleted; mixed growth of mossy Li and dendrites during partial depletion; and dendrite growth after full depletion. The three-stage behavior can be well-explained by heterogeneous Li$^+$ depletion on the Li surface. Without SRS microscopy, a clear correlation between the Li$^+$ concentration and dendrite growth cannot be seen and validated. We also discovered that the Li$_3$PO$_4$-based artificial SEI layer helps suppress dendrite growth regardless of ion depletion, which demonstrates the effectiveness of this approach. On the other hand, TBA-PF$_6$ is very effective as an electrolyte additive in suppressing uneven Li growth in stage I but cannot prevent dendritic Li from forming in stage III. Our results suggest that the abilities of electrolyte additives and other suppression methods need to be examined in the different stages of Li growth, especially at a high current density, i.e., where stage III can occur. This study provides insight into the Li dendrite growth

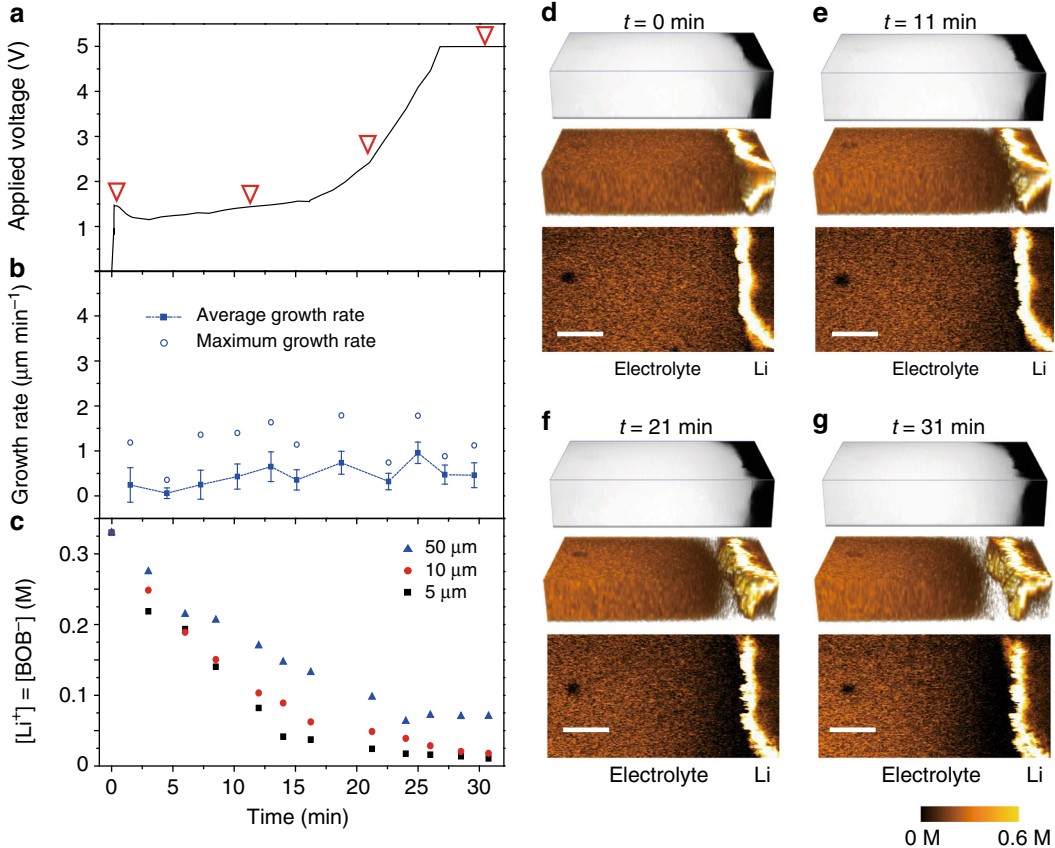

**Fig. 5** SRS analysis of a $Li_3PO_4$-coated Li electrode. **a** Voltage profile of a $Li_3PO_4$-coated Li–Li cell showing the voltage change vs. time. **b** Average growth rate and standard deviation versus time (same sizes as above). **c** Concentration change at 5 µm, 10 µm and 50 µm away from the electrode surface. **d**–**g** Correspond to 0 min, 11 min, 21 min and 31 min, respectively. Each time set contains a 3D image, 3D SRS image, and representative 2D SRS image from top to bottom. The depletion region in all 3D SRS images is represented by a transparent white area to enhance the visual difference

mechanism and design principles for dendrite-free Li metal batteries and demonstrates the power of SRS microscopy for materials and energy studies. Given its excellent resolutions, SRS microscopy can be further applied to investigate relevant topics in materials science, such as ion transport/intercalation inhomogeneity, dynamic ion movement in confined channels, material separation, and catalytic process products. It will guide the design of high-performance materials and systems.

## Methods

**Gel electrolyte preparation.** Gel electrolyte was used in the study as it minimizes convection and electro-osmotic effect in our customized glass cells. To prepare the gel electrolyte, 0.33 M Lithium bis(oxalato)borate (LiBOB, Sigma-Aldrich) in Tetraethylene glycol dimethyl ether (TEGDME, Sigma-Aldrich) electrolyte was prepared first. Then LiBOB/TEGDME solution, PVdF-HFP (Kynar Flex 2801), and DME were mixed at a weight ratio of 7:2:14 and stirred overnight to form a transparent solution. Then the solution was dropped between two lithium metal electrodes inside glovebox and left for four hours to let DME evaporate. The final composition of the electrolyte is 0.33 M LiBOB gel electrolyte in which PVdF-HFP is 22 wt%.

**Lithium-lithium symmetric cell preparation.** First, four layers of Kapton tapes were laminated onto a glass slide, and two chambers of (~1 × 1 cm) and a channel (~1 × 6 mm) which connects two chambers were made by cutting and removing Kapton tapes. Then lithium foil with the desired size was placed to two ends of the chamber so that the foil dimension fitted the chamber size. Then the precursor of the gel electrolyte was added and let DME solvent evaporate. Afterward, Cu foils were placed at the two ends of lithium for external electrical contact. A glass cover is placed on top of the Li/Li cell, and epoxy is used to seal the cell. The whole process is done in a glove box with $O_2$ and water level < 0.1 p.p.m. $Li_3PO_4$-based artificial SEI was created by soaking Li metal in DMSO with 0.4% wt $H_3PO_4$ solution for 2 min.

**Electrochemical characterization.** All tests were performed on Land tester which has 1 mA / 5 V range with a resolution of 1 µA and 1 mV. Impedance measurements for calculating ionic conductivity and diffusion coefficient were performed using a potentiostat (Biologic VMP3).

**Stimulated and spontaneous Raman scattering microscopy setup.** Detailed SRS setup is reported previously by the authors[45]. Briefly, picoEMERALD laser (Applied Physics & Electronics, Inc.) provides an output pulse train at 1064 nm (6-ps pulse width and 80 MHz repetition rate), serving as the Stokes beam. The frequency doubled beam at 532 nm is used to synchronously seed a picosecond optical parametric oscillator (OPO) to produce a mode-locked pulse train with 5~6 ps pulse width. A built-in electro-optic modulator modulates the intensity of the 1,064-nm Stokes beam at 8 MHz. The pump beam is then spatially and temporally overlapped with the Stokes beam by using a dichroic mirror inside picoEMERALD and then coupled into an inverted multiphoton laser-scanning microscope (FV1200MPE, Olympus) optimized for near-IR throughput. A ×25 water objective (XLPlan N, 1.05 numerical aperture (NA), MP, Olympus) with the high near-IR transmission is used for imaging. The forward going pump and Stokes beams after passing through the sample are collected in transmission with a high N.A. condenser lens (oil immersion, 1.4 N.A., Olympus). A telescope is then used to image the scanning mirrors onto a large area (10 × 10 mm) Si photodiode (FDS1010, Thorlabs) to descan beam motion during laser scanning. The photodiode is reverse biased by 64 V from a DC power supply. A high O.D. bandpass filter (890/220 CARS, Chroma Technology) is used to block the Stokes beam and to transmit the pump beam only. The output current of the photodiode is electronically pre-filtered by an 8-MHz band-pass filter (KR 2724, KR Electronics). It was then fed into a lock-in amplifier (HF2LI, Zurich instrument), terminated with 50 Ω to demodulate the stimulated Raman loss signal. The in-phase X-output from lock-in amplifier was fed back into the analog interface box (FV10-ANALOG) of the microscope. For all imaging, 256 × 256 pixels were acquired with a 2 µs of pixel dwell time (0.13 s per frame, 1.2 µs time constant from the lock-in amplifier) for laser scanning, averaged over 4 or 16 frames to reduce noise and minimize the heat effect. Laser powers (after objective) used for imaging were: 24 mW for the pump beam and 50 mW (or 24 mW for 16 depths *z* scan) for Stokes beam. The spontaneous Raman spectra were acquired by a laser confocal Raman microscope

(Xplora, Horiba Jobin Yvon). A 532-nm diode laser (12 mW, after the microscope objective) was used to excite the sample through a 50× air objective (MPlan N, 0.75 N.A., Olympus). The acquisition time was 14 s for each spectrum with the LabSpec 6 software.

**Image processing and data extraction**. 3D images in Figs. 3, 5 in main text were constructed by ImageJ without further treatment. In Fig. 3, To make dendrite more visually distinct, the image is elongated along $x$-direction by 66.7%. To obtain images with jet color bar in Fig. 4, Supplementary Fig.10, Fig. 12, Fig. 14 and Fig. 16, original 4 frame-averaged 2D images (8 μs per pixel) were further denoised by kernel smoothing, to reduce noise from ~20 mM to < 10 mM. An example of kernel smoothing is shown in Supplementary Fig. 9. To smooth an image, the kernel smoothing method extracts every pixel and builds a probability distribution function (PDF) using the surrounding sample data. Then the kernel distribution sums the smoothing functions for each set to produce a smooth, continuous PDF curve. The code of Kernel smoothing can be found at: https://github.com/aciditeam/matlab-ts/blob/master/ksrmv.m.

Detail about Kernel smoothing can be found at Matlab website: https://www.mathworks.com/help/stats/kernel-distribution.html

To extract quantitative concentration, the average SRS intensity in electrolyte before applying current is assigned as the initial concentration (0.33 M), as Li+ concentration should be uniform before the current is applied. Intensity for 0 M is determined by average signal in another SRS image of 1877 cm$^{-1}$ at the same height ($z$). Then the observed intensity can be converted to local Li+ concentration profile based on the linear relation between SRS intensity and Li+ concentration (Fig. 1d). In Fig. 3g, the concentration at each point between two electrodes along $x$-direction is calculated from the average concentration of all points with the same coordinate $x$. A five-point moving average is also applied along the $x$-direction. The dendrite growth rates are calculated by dividing the local growth length by the intervening period between two adjacent images.

**Data availability**. The data that support the findings of this study are available from the corresponding author upon reasonable request.

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

## Acknowledgements

We acknowledge seed funding support from Columbia University's Research Initiatives in Science & Engineering competition, started in 2004 to trigger high-risk, high-reward, and innovative collaborations in the basic sciences, engineering, and medicine. Y.Y. acknowledges support from startup funding by Columbia University. W.M. acknowledges support from the US Army Research Office (W911NF-12-1-0594), NIH Director's New Innovator Award (1DP2EB016573) and R01 (EB020892), and the Camille and Henry Dreyfus Foundation. Z.L and L.-Q.C. acknowledge the support from the Department of Energy, Office of Energy Efficiency and Renewable Energy (EERE), under the Award (DE-EE0007803). Y.Y and Q.C. also want to thank Prof. Alan West at Columbia University for his kind help.

## Author contributions

Y.Y., W.M., Q.C., and L.W. conceived the idea and designed the experiments. Q.C. and L. W. performed all the experiments and measurements. Z.L, Z.S. and L-Q. C. performed simulations. N.N., B.Z., W.X., M.C., and Y.M. helped prepare and perform experiments. All authors discussed the results. Q.C., L.W., Y.Y., W.M. and Z.L. wrote the paper with the input from all authors.

## Additional information

**Competing interests:** The authors declare no competing interests.

