## [Peer Review File · Nature Communications]

Reviewers' comments:

Reviewer #1 (Remarks to the Author):

This is a potentially exciting approach to gaining more understanding of the dendrite growth process, and I think that in principle it provides important information about local heterogeneity in electrolyte, something that hasn't been measured.

However, I have 2 serious concerns about the experimental technique, the first of which causes me to wonder about the validity of their conclusions.

1. Most critical, the authors claim that they can determine the Li^+ concentration by measuring the BOB^- concentration, "since electro-neutrality requires that the local concentration of Li^+ and BOB^- must be equal to each other." While this statement appears plausible, it is in general not true in battery electrolytes. That's because there are usually numerous ion complexes that are also present at significant concentrations. These complexes "may be designated as solvent-separated ion pair (SSIP), contact ion pair (CIP) or aggregate (AGG) solvates in which the anions are coordinated to zero, one and two or more Li^+ cations, respectively." [J. Electrochem. Soc. 2012 volume 159, issue 9, A1489-A1500]. There is a substantial literature on these species for a variety of solute concentrations in Li ion battery electrolytes. The situation is still further complicated by the fact that the spectra of these ions change--in both frequency and intensity--when complexed, so I don't know whether their BOB^- measurements are for free BOB^- or for solvated BOB^- (CIPs or AGGs involving BOB^-) ions or both. Because of all of these issues, it's not clear what the authors are actually measuring and whether/how this situation affects the interpretation of their dendrite data. On the other hand, they are certainly measuring something that is closely related to the number of positively charged Li^+ ions, since, for example, they get plausible diffusion coefficients. [Unfortunately, there is a rather wide disagreement in the literature about just what the diffusion coefficient values should be, so a "plausible" value doesn't necessarily mean a "correct" value.] The authors need to either refute the argument given above or recast the paper to argue that they can still learn about dendrite formation even if it's not clear exactly what they are measuring.

2. The authors claim an advance in that they are using SRS, but I am not convinced of the value of using SRS for such measurements. In particular, spontaneous Raman was used to get more or less equivalent information, as claimed in their reference 39. The present authors state, incorrectly, that the spontaneous Raman technique "has a rather poor temporal resolution (~10 minutes / frame), far from being sufficient to follow the rapidly changing concentration of electrolyte." In fact, Raman signals in condensed materials are often very strong. Reference 39 states that "Spectra were collected using a single 5 second integration," which means that its time resolution is similar to that of SRS. In addition, the spontaneous Raman measurements have at least 2 advantages. First, the technique is so simple that any student of freshman chemistry with a Raman instrument can take high quality spectra, which makes it a highly accessible technique. By comparison, SRS is relatively complex. Second, it is clear from the spectroscopy what species is being observed in Reference 39, making interpretation more straightforward, although even in that case the situation is not necessarily simple. It is true that SRS is an intrinsically 3D technique, while the spontaneous Raman technique is nominally only a 2D technique. However, because of the confocal nature of the spontaneous Raman experiment, the depth into the electrolyte of the observation point is controlled by adjusting the laser beam focus (https://en.wikipedia.org/wiki/Confocal_microscopy). The authors should either justify their use of SRS compared to spontaneous Raman or else state that SRS is simply an alternative to the spontaneous Raman technique that might be useful in some cases.

By the way, it is my understanding that the correct term is "operando," not "in-operando."

Reviewer #2 (Remarks to the Author):

This article by Cheng et al describes the application of SRS imaging to monitoring lithium electrodeposition. This work has obvious relevance to understanding defect origins in batteries, and the electrodeposition experiments and interpretation seem very reasonable. It is concerning that the conclusions are based off of only a few devices, rather than being a statistically reasonable sample size. However my primary concern is that authors have not sufficiently demonstrated that their electroneutrality argument should hold under all conditions of their experiments. This is a crucial point, as their experiments are completely insensitive to lithium ions directly. The authors need to perform a number of control experiments and more readily acknowledge this limitation of SRS before this article should be considered for publication. Specifically:

1. Every result presented here relies on the assumption that monitoring the counterion solution is equivalent to monitoring the lithium ions directly. The authors are clearly aware of this point, based on their discussion of electroneutrality in the SI. However, given the importance of this point to the entire paper, Comsol modeling and application of Poisson's equation is not sufficient to prove that this holds for dynamic chemical systems with nano and microstructured electrodes. The authors need to do some key control experiments to prove that their assumption is valid, and provide convincing experimental SRS evidence that the BOB⁻ concentration does accurately track the Li⁺ concentration. A quantitative value for the sensitivity extrapolated to lithium ion concentrations is also needed.
2. Related to point 1, the paper reads as if the authors are directly measuring Li⁺ concentrations (see title, y axis of Figure 1f, 2c, 2f, etc). Particularly for readers not familiar with Raman spectroscopy, this could lead to serious confusion. The authors should be straightforward and acknowledge more directly that there are no lithium ion signatures in the Raman spectrum throughout the entire manuscript.

Summary of changes:

At Page 5 in the manuscript: "More discussions on the effect of microelectrodes, ion/solvent interactions on electroneutrality can be found in Supplementary note 2, 3 & 4 and Supplementary Fig. 2, 3 & 4."

At Page 6 in the manuscript: "It should be noticed that all $[\text{Li}^+]$ in this study is measured based on $[\text{BOB}^-]$, and their difference is much smaller than resolution of SRS in the electrolyte (Supplementary note 2)."

New Supplementary note 2 & 3 and Supplementary Fig. 2&3 for validation of electroneutrality including experiments, simulations and modeling.

New Supplementary note 4 and Supplementary Fig. 4 for the salt-solvent interaction and linearity tests.

Answers to Reviewers' Comments:

Reviewer #1 (Remarks to the Author):

This is a potentially exciting approach to gaining more understanding of the dendrite growth process, and I think that in principle it provides important information about local heterogeneity in electrolyte, something that hasn't been measured.

However, I have 2 serious concerns about the experimental technique, the first of which causes me to wonder about the validity of their conclusions.

*Question 1: 1. Most critical, the authors claim that they can determine the Li^+ concentration by measuring the BOB^- concentration, "since electro-neutrality requires that the local concentration of Li^+ and BOB^- must be equal to each other." While this statement appears plausible, **it is in general not true in battery electrolytes. That's because there are usually numerous ion complexes that are also present at significant concentrations.** These complexes "may be designated as solvent-separated ion pair (SSIP), contact ion pair (CIP) or aggregate (AGG) solvates in which the anions are coordinated to zero, one and two or more Li^+ cations, respectively." [J. Electrochem. Soc. 2012 volume 159, issue 9, A1489-A1500]. There is a substantial literature on these species for a variety of solute concentrations in Li ion battery*

electrolytes. The situation is still further complicated by the fact that the spectra of these ions change--in both frequency and intensity--when complexed, so I don't know whether their BOB⁻ measurements are for free BOB⁻ or for solvated BOB⁻ (CIPs or AGGs involving BOB⁻) ions or both. Because of all of these issues, it's not clear what the authors are actually measuring and whether/how this situation affects the interpretation of their dendrite data. On the other hand, they are certainly measuring something that is closely related to the number of positively charged Li⁺ ions, since, for example, they get plausible diffusion coefficients. [Unfortunately, there is a rather wide disagreement in the literature about just what the diffusion coefficient values should be, so a "plausible" value doesn't necessarily mean a "correct" value.] The authors need to either refute the argument given above or recast the paper to argue that they can still learn about dendrite formation even if it's not clear exactly what they are measuring.

Answer: We thank the Reviewer 1 for the positive feedback. We think that this question has three key points: 1. whether electroneutrality is valid if there are SSIP, CIP and AGG. 2. If electroneutrality is valid, does it mean Li⁺ equal to anion concentration as SSIP/CIP/AGG exist? 3. Even if [Li⁺] = [anion⁻], could Raman signal quantitatively reflect anion concentration, and thus Li⁺, when interactions above present? We will answer them one by one. The following discussion is also added to supplementary information as Supplementary Note 4.

First, let us consider the case that there is only one salt, Li⁺ and one anion (e.g. BOB⁻).

1. Whether electroneutrality is valid if there are SSIP, CIP and AGG.

Let us assume besides free Li⁺ and free BOB⁻, the electrolyte contains a complex consisting of $n \cdot \text{Li}^+ + m \cdot \text{BOB}^- + l \cdot \text{neutral solvent molecule}$ due to any interaction above ($n, m, l = 0, 1, 2, 3, \dots$). Their concentrations are C_+ , C_- and C_{com} , respectively. Obviously their charge numbers are 1, -1 and $n-m$, respectively. Poisson equation shows

$$\nabla^2 \phi = -\nabla E = -\frac{\rho}{\epsilon_r \epsilon_0} = -F * \Sigma(C_+ + C_- + (n - m)C_{com})/\epsilon_r \epsilon_0.$$

The local electrical field can be calculated by $E = j/\sigma$ (current density/ionic conductivity). As our resolution is 10 mM, we consider 1 mM Li salt as an extreme case, whose σ is $\sim 10^{-5}$ S/cm. Assume a high j of 10 mA/cm², then E is 10 mA/cm² / 10^{-5} S/cm = 10^5 V/m. Let us assume E

jumps from 0 to 10^5 V/m within just 1 μm , which is much shorter than reality. Then ∇E is $\sim 10^5 \text{V/m} / 1 \mu\text{m} = 10^{11} \text{V/m}^2$.

If electrolyte dielectric constant ϵ_r is 100, ρ is only 10^2C/m^3 since ϵ_0 is $\sim 10^{-11} \text{F/m}$, which means that $C_+ + C_- + (n - m)C_{com}$ is only 10^{-6}mol/L . Even if the change of 10^5V/m happens within 10 nm, $C_+ + C_- + (n - m)C_{com}$ is 10^{-4}mol/L , **much less than our resolution of 0.01 mol/L. Therefore, $C_+ + C_- + (n - m)C_{com}$ can still be considered as 0.**

We have provided more detailed experiments in the supporting experiments to support the electroneutrality. (**Supporting Note 2&3, Supporting Figure 2&3**)

2. If electroneutrality is valid, is Li^+ equal to anion concentration when SSIP/CIP/AGG exist?

In such an electrolyte, the local total $[\text{Li}^+]$ is $C_+ + n * C_{com}$, the sum of free Li^+ and those in the complex. Similarly, local total anion concentration is $C_- + m * C_{com}$. Therefore, their difference is $C_+ - C_- + (n - m)C_{com}$, which is “zero” as shown above. **So local equivalency of $[\text{Li}^+]$ and anion concentration is not violated due to any ion-ion and ion-solvent interaction.** Or in other word, the difference is less than 0.1 mM even in the extreme case.

3. If electroneutrality is valid, is Raman signal proportional to anion concentration, and thus Li^+ , as Raman peaks may split or distorted due to ion interaction?

The short answer is that we don't observe such nonlinearity in LiBOB up to solubility and in LiTFSI up to 2 M, respectively, which may be a result of large size of anion and weak interaction among them. The long answer is two folds as below:

a. We have experimentally observed high linearity in **anion concentration** vs. Raman intensity in LiBOB/TEGDME, LiTFSI/TEGDME, and LiTFSI/ DMC up to high concentrations (Figure R1). In BOB, it is 0-0.5 M as the solubility is $\sim 0.6 \text{M}$, and it is 0-2 M for LiTFSI in TEGDME or in DMC. The Raman intensity is for a given wavenumber instead of peak integration, to make it consistent with SRS measurement. Both correlation coefficients are higher than 0.999. We indeed need to identify these modes, to see whether CIP/AGG is involved, but no matter whether it is for free or with interaction, experimental Raman signal is linear with anion concentration in reasonable range for battery electrolyte, as shown in Figure R1. Especially as we are studying low ion concentration in depletion, linearity should be general in this range.

Regarding to the JES 2012 paper the reviewer cite, Figure 3 in the paper shows, linearity exists for salt molar fraction < 0.15 . This corresponds to 2.0 M, 2.5 M and 3.0 M for LiTFSI, LiPF₆ and LiClO₄, respectively. Therefore, strong interaction to cause nonlinearity only occurs for really high concentration, which is not common in battery systems.

Figure R1. (a) Spontaneous Raman of LiTFSI in TEGDME at different concentrations. (b) Linear relation between Raman intensity and LiTFSI concentration at three wavenumbers. (c) Spontaneous Raman of LiBOB in TEGDME at different concentrations. (d) Linear relation between Raman intensity and LiBOB

concentrations at three wavenumbers. (e) Spontaneous Raman of LiTFSI in DMC at different concentrations. (f) Linear relation between Raman intensity and LiTFSI concentration at three wavenumbers. All peaks correspond to anions, and the Raman intensity in each peak in (b), (d) and (f) is for a specific wavenumber, but not integral of the peak, to mimic intensity of single wavenumber detection in our Stimulated Raman scattering microscopy.

b. Whether Raman intensity of direct **Li⁺-solvent interaction** is linear with Li⁺ concentration or not. A good sign is that Raman signal of Li-ether interaction shows a linear dependence up to PEO/LiTFSI = 10:1 (~1.7 M LiTFSI, Fig. 2).¹ Above 1.7 M, Raman intensity still increases monotonically with Li⁺, which can also satisfy quantitative analysis.¹ In JES 2012 paper the reviewer mentioned, from figure 3, the linearity exists for salt molar fraction < 0.15 in acetonitrile. This corresponds to 2.0 M, 2.5 M and 3.0 M for LiTFSI, LiPF₆ and LiClO₄, respectively. We also examine the linearity using LiBOB / (TEGDME:EC v/ 7:3) in which [Li⁺] can be tracked by the Li-EC interaction at 725 cm⁻¹. As shown in Figure R4 on page 12 below, which is also Supplementary Figure 3a, the intensities of Li⁺-EC have very good linearity with Li⁺ concentration.

Editorial Note: Figure R2 in this Peer Review File has been amended to remove third-party material where no permission to publish could be obtained. See figure in J. Electrochem. Soc., Vol. 145, No. 9, 3034 1998, ¹.

Second, we will discuss how to deal with multiple salt conditions:

1. Although our current setup can only track a single wavenumber at a time, it is possible to track multiple ones by adding more optical parameter oscillator (OPO) inside. For example, Minbiao Ji reported simultaneous dual wavenumber detection in 2017.² Moreover, the time for OPO to switch to another wavenumber in our setup can be as fast as only 2-5 seconds, which can enable quasi-simultaneous imaging of multiple wavenumbers.

2. Even with a single wavenumber, as transference numbers of different ions can be determined prior to experiment, which are especially near constant as low concentration, we can derive total anion concentration first from transference number, modeling and Raman of a single anion. Then determine total Li^+ concentration as it equals total anion.

As a summary of answer to question 1, **Electroneutrality is valid in battery electrolyte. The difference between $[\text{Li}^+]$ and $[\text{anion}]$ is much less than resolution of SRS (1-10 mM), no matter whether interaction exists.** More discussions on electroneutrality will be presented in answer to reviewer 2.

Question 2. The authors claim an advance in that they are using SRS, but I am not convinced of the value of using SRS for such measurements. In particular, spontaneous Raman was used to get more or less equivalent information, as claimed in their reference 39. The present authors state, incorrectly, that the spontaneous Raman technique "has a rather poor temporal resolution (~10 minutes / frame), far from being sufficient to follow the rapidly changing concentration of electrolyte." In fact, Raman signals in condensed materials are often very strong. Reference 39 states that "Spectra were collected using a single 5 second integration," which means that its time resolution is similar to that of SRS. In addition, the spontaneous Raman measurements have at least 2 advantages. First, the technique is so simple that any student of freshman chemistry with a Raman instrument can take high quality spectra, which makes it a highly accessible technique. By comparison, SRS is relatively complex. Second, it is clear from the spectroscopy

what species is being observed in Reference 39, making interpretation more straightforward, although even in that case the situation is not necessarily simple. It is true that SRS is an intrinsically 3D technique, while the spontaneous Raman technique is nominally only a 2D technique. However, because of the confocal nature of the spontaneous Raman experiment, the depth into the electrolyte of the observation point is controlled by adjusting the laser beam focus (https://en.wikipedia.org/wiki/Confocal_microscopy). The authors should either justify their use of SRS compared to spontaneous Raman or else state that SRS is simply an alternative to the spontaneous Raman technique that might be useful in some cases.

Answer: we have read ref 39 carefully, and our understanding is that the spectrum at a **single spot (pixel)** takes 5 seconds in ref 39, while ours is ~5 s for a **whole 256X256X5 stack** with 16 microsecond (2 μ s dwelling time **X 8 average**) for a single spot (pixel). Therefore, besides deriving diffusion coefficient from ~20 spots, which takes ~100 s in ref 39, we can also capture 2D/3D heterogeneity and dynamics.

Reviewer #2 (Remarks to the Author):

This article by Cheng et al describes the application of SRS imaging to monitoring lithium electrodeposition. This work has obvious relevance to understanding defect origins in batteries, and the electrodeposition experiments and interpretation seem very reasonable. It is concerning that the conclusions are based off of only a few devices, rather than being a statistically reasonable sample size.

Answer: We thank the reviewer for the positive feedback. First, we collect information from all the points of electrode surface, and the data listed in the paper are analyzed statistically; second, we have run dozens of devices which can generate similar conclusion behind each figure. Limited by the length of manuscripts, only a few representative results can be displayed. Here we provide another set of data with a higher current density (2 mA/cm²) which show that the same conclusion can stand at a different condition.

Figure R3. 2D overlapped images at three representative moments, a-c: 3/8 min (Stage I / non depletion), d-f: 15/20 min (Stage II / parital depletion), g-i: 35/40 min (Stage III / full depletion) are picked to show local inhomogeneity in Li⁺ concentration and dendrite growth. The first column exhibits the ionic concentration profile and solid Li electrode region at t_1 (a / 3 min, d / 15 min & g / 35 min). The boundaries of lithium electrode at t_1 and t_2 are labeled by red and white dash lines, respectively. The second column exhibits Li⁺ concentration at t_2 (b / 8 min, e / 20 min & h / 40 min). The gray region at bottom indicates the difference of solid Li electrode between t_1 and t_2 . The white arrows represent the dendrite growth direction and rate. The Li⁺ concentration in the electrolyte is represented by jet color bar to amplify the visual contrast. Color bar is at right. Arrows in electrolyte represent concentration gradient in the electrolyte. The third column (c, f, i) shows

the relationship between dendrite growth rate (black lines) and $[Li^+]_{10\mu m}$ versus location at t1 (blue lines) / t2 (red lines), respectively.

Table 1. Correlation between local Li growth rate and heterogeneity in Li^+ concentration.

		Non Depletion (3/8 min)	Partial Depletion (15/20 min)	Full Depletion (35/40 min)	
Experimental Results	v ($\mu m/min$)	1.35 ± 0.94	1.94 ± 1.36	4.42 ± 3.46	
	$[Li^+]_{10\mu m}$ (mM)	t1	157 ± 16	55 ± 16	43 ± 26
		t2	101 ± 13	54 ± 14	41 ± 24
	R of v and $[Li^+]_{10\mu m}$ in Figure R3	t1	0.09	0.15	0.42
		t2	0.22	0.48	0.66

The relation between $[Li^+]_{10\mu m}$ and local Li growth rate is shown in Figure R3 while analytical results (dendrite growth rate, $[Li^+]_{10\mu m}$ (mM) and correlation coefficients) are listed in Table.1. First of all, it is clear that the lithium deposition still shows three stages, slow lithium deposition (Non depletion), expedited lithium deposition and dendrite initiation (Partial depletion) and erupted dendrite growth (Full depletion).

Similar phenomena are also observed here with results in our manuscript: First, high and increasing correlation between the spatial distribution of v and $[Li^+]_{10\mu m}$ is observed. correlation coefficient (R) of v and $[Li^+]_{10\mu m}$ at t2 increases from 0.22 at stage I to 0.48 at stage II and 0.66 at stage III. In our correlation analysis, the number of data point is 166, the corresponding R value to reach a confidence level of 0.01 and 0.05 are 0.22 and 0.16, respectively. Therefore, 0.48 and 0.66 indicate significant correlation between v and $[Li^+]_{10\mu m}$ in partial depletion and full depletion states.

Second, R of v and $[Li^+]_{10\mu m}$ at t2 is always higher than that at t1 for all three stages, and finally, increasing fluctuation in both v and $[Li^+]_{10\mu m}$ from stage I to III. These phenomena support the same conclusion from the manuscript.

However my primary concern is that authors have not sufficiently demonstrated that their electroneutrality argument should hold under all conditions of their experiments. This is a crucial point, as their experiments are completely insensitive to lithium ions directly. The authors need to perform a number of control experiments and more readily acknowledge this limitation of SRS before this article should be considered for publication. Specifically:

1. Every result presented here relies on the assumption that monitoring the counterion solution is equivalent to monitoring the lithium ions directly. The authors are clearly aware of this point, based on their discussion of electroneutrality in the SI. However, given the importance of this point to the entire paper, Comsol modeling and application of Poisson's equation is not sufficient to prove that this holds for dynamic chemical systems with nano and microstructured electrodes. The authors need to do some key control experiments to prove that their assumption is valid, and provide convincing experimental SRS evidence that the BOB⁻ concentration does accurately track the Li⁺ concentration. A quantitative value for the sensitivity extrapolated to lithium ion concentrations is also needed.

Answer: We are sorry for not explaining electroneutrality comprehensively enough and appreciate the reviewer's comments to help clarify this point and improve our manuscript. We will answer reviewer's question from four aspects: **1. experimental proof; 2. analytical solutions; 3. COMSOL Simulation on microelectrode; and 4. Simulation based on experimental data.** The following discussions are also added to Supplementary Note 2&3 and Supplementary Figure 2&3.

Before discussing these supportive arguments, we would like to first clarify the meaning of electroneutrality in electrochemistry and this paper. We don't claim that the difference between [BOB⁻] and [Li⁺] is absolutely zero, but **the difference is very small, and it can be neglected at given resolutions of SRS microscopy.** As SRS has resolution of 1-10 mM and ~500 nm in our experiment, and our measurement is at least 1 μm away from electrolyte/electrode interface, the electroneutrality means that at scale of ~100 nm, the difference between [BOB⁻] and [Li⁺] is much less than 1 mM. Therefore, such difference can be neglected in our analysis. This is the foundation of following discussions.

1. Experimental Proof

We don't find any established experimental method in literature to justify electroneutrality. Therefore, we develop an approach that is most solid based on our experience and discussions with electrochemists. We hope that the results can satisfy reviewers, and we appreciate if reviewers can provide suggestions on experiments if more validations are needed.

As the SRS setup at Columbia can only detect wavenumber higher than 1000 cm^{-1} , we cannot directly see Li^+ ion based on Li-electrolyte interaction. Therefore, we track both Li^+ and BOB^- near electrode surface by spontaneous Raman to derive their local concentration. The electrolyte used in this experiment is $\text{LiBOB} / (\text{TEGDME}:\text{EC v/ 7:3})$ instead of $\text{LiBOB}/\text{TEGDME}$, so that $[\text{Li}^+]$ can be tracked by the Li-EC interaction at 725 cm^{-1} , and BOB^- at 1830 cm^{-1} (Figure R4a).

As shown in Figure R4b, the intensities at the designated wavenumbers have very good linearity with concentration, so that the derived $[\text{Li}^+]$ and $[\text{BOB}^-]$ have high accuracy. The standard deviation is determined to be 8.3 mM for Li^+ -EC solvation and 5.8 mM for BOB^- for a 14 second accumulation per point. It should be noted that the accumulation time is 10^6 longer than SRS to get similar chemical resolution.

Figure R4. (a) The Raman spectrum of $\text{LiBOB} / (\text{TEGDME}:\text{EC v/ 7:3})$ electrolyte with a concentration from 0 M to 0.4 M. The accumulation time is 14 seconds without average for each spot. (b) The plot of counts at designated wave number (725 cm^{-1} for Lithium solvation and 1825 cm^{-1} for BOB^-) versus concentration of LiBOB .

Then we built the same lithium-lithium electrolytic cell as Figure 1c in the main text to probe local ion concentration under spontaneous Raman (XploRA One by HORIBA). Then this

electrolytic cell was tested under variant current densities (from the 0.5 mA/cm² to 1.5 mA/cm², Figure R5A) until ions at electrode surface are fully depleted together with lithium dendrite growth. The changing current density can also help verify the electroneutrality under different electrical field.

During the cell operation, lithium is deposited as dendrite and gradually approach the Raman spot which has a size ~10 μm. We find that when the laser directly shines at lithium, severe signal loss and distortion will occur, which may be due to interaction with SEI and lithium. Therefore, we only approach to ~ 10 μm away from the lithium surface, which is similar with our paper (Figure R5B).

Figure R5. (a) The voltage and current curve versus time plot in a lithium- lithium electrolytic cell. (b) The optical image of electrode at the beginning and at the end of lithium electrodeposition. After the deposition, there are large amount of dendrites formed on the electrode surface and the colored squares in the optical images show the positions where Raman spectra were taken.

After converting Raman intensity to ion concentration, we can clearly see that [Li⁺] and [BOB⁻] are reduced due to ion depletion (Figure R6A&B). Their absolute concentrations synchronize with each other, so that they appear to be the same as each other and electroneutrality is valid.

Figure R6. (a) Raman Spectra taken from the start of deposition (0 min) to the end of the deposition (65 min). The intensity of both peak decreases over time. (b) The concentration change of Li^+ and BOB^- versus time.

To further answer this question more quantitatively, we performed the hypothesis testing in statistics. Data at 50 minutes were used as an example here. For cations, $X_C \sim N(\mu_C, \sigma_C^2)$, X_C is the measured cation concentration 0.0153 M and σ_C is the standard deviation of cation concentration 0.0083 M. The same go with anions, $X_A \sim N(\mu_A, \sigma_A^2)$ while $X_A = 0.0146$ M and $\sigma_A = 0.0064$ M. We want to test whether the mean concentration for cations μ_C and anions μ_A are consistent. The null hypothesis is $\mu_C = \mu_A$ with a significance level of 1%.

$X_C - X_A$ is normal distributed with a mean of $\mu_C - \mu_A$ and standard deviation of $\sqrt{\sigma_C^2 + \sigma_A^2}$.

For hypothesis $\mu_C - \mu_A = 0$,

$$\frac{X_C - X_A - 0}{\sqrt{\sigma_C^2 + \sigma_A^2}} \sim N(0, 1) \quad (\text{Equation R1})$$

The test statistic is 0.074, far smaller than $Z_{0.005}$ ($Z_{0.005} = 2.58$, the value can be found in Z table), so the null hypothesis is accepted. Other experimental results (50 min - 65 min) with test statistics (-0.363, 0.082, -0.216, 0.148) all fall in the range from $-Z_{0.005}$ to $Z_{0.005}$.

It is worth noticing that the sum of the square of each time also follows chi-square distribution. The chi-square test statistics is $\sum \left(\frac{X_C - X_A}{\sqrt{\sigma_C^2 + \sigma_A^2}} \right)^2 = 0.182$, which is much smaller than the chi-square table at degree of freedom of 5 at significance level of 5% (16.75). We can conclude that the concentration of cation and anion are equal in the whole experiment.

2. Analytical Solutions

The effectiveness of electroneutrality on microelectrodes was actually analyzed by Prof. Henry White at University of Utah³. Although the title is “violation of electroneutrality”, it actually shows that the difference between cation and anion concentration ($\ll 1$ mM) is much less than our resolution, and **his “violation” simply means that when electrode is nano-sized, the concentration difference is μ M-level, which is much larger than that of flat surface (e.g. nM or below)**. In equation 9 of reference 3, Henry White showed that under voltammetric response and outside the Debye length, the local net charge in electrolyte (Σ_{error}) near a spherical microelectrode satisfies

$$-r_0^2 \frac{\epsilon \epsilon_0}{F} \nabla^2 \phi = 18.4 \text{ nm}^2 \text{ mM} \left[\frac{r}{r_0} + \frac{2\gamma(r/r_0)^2}{I/I_l} \right]^2 \equiv r_0^2 (c_+ - c_-) \quad (\text{Equation R2})$$

Where r_0 is the radius of spherical microelectrode, F is the Faraday constant (96485 C/mol), ϵ is relative permittivity ($\epsilon=7.8$ for TEGDME)⁴ and ϵ_0 is vacuum permittivity ($8.85 \cdot 10^{-12}$ F*m⁻¹), Φ is the potential difference between the position r in the solution and the bulk of the solution, γ represents the ratio of supporting electrolyte to reactant concentration, I/I_l is current after being normalized by limiting current and, c_+ / c_- are concentrations of all cations and anions, respectively. **It should be noted that 18.4 represents value for TEGDME with ϵ of 7.8 instead of 78 in water in the original paper.**

Therefore, the maximum charge difference occurs at $\gamma = 0$, which means no supporting electrolyte to shield electrical field. This is also the same as our experimental condition, where the electrolyte is binary without supporting electrolyte ($\gamma = 0$). In this case, the second term in the square brackets is zero and the equation can be simplified to

$$18.4(\text{nm}^2 \text{ mM})r^{-2} \equiv (c_+ - c_-) \quad (\text{Equation R3})$$

Based on equation 3, the difference between $[\text{Li}^+]$ and $[\text{BOB}^-]$ is purely determined by the distance to the center of the microelectrode. Therefore, r is much smaller in microelectrode than bulk electrode. Even if the electrode size is 10 nm, and the SRS measurement is 100 nm away from the electrode/electrolyte interface, $c_+ - c_- = 18.4 / (110^2) = 1.52 \cdot 10^{-3}$ mM, which is much less than SRS resolution. Even for water, it is only $1.52 \cdot 10^{-2}$ mM. In addition, the typical Debye length is ~ 5 nm at the concentration of 1 mM, so 100 nm satisfy the prerequisite of

outside Debye region. Hence, the analytical studies above prove that electroneutrality is still valid even on microelectrodes.

3. Simulation on microelectrode.

To further validate the analytical solution that $C_+ - C_-$ is less than 1 mM, COMSOL simulation on microelectrode is performed. The parameters used are: Diffusion coefficient is $5 \times 10^{-7} \text{ cm}^2/\text{s}$, current density is $2 \text{ mA}/\text{cm}^2$, and electrode distance is $100 \text{ }\mu\text{m}$. The electrode is designed to have a single tip with a **width of 10 nm** and a length of $2 \text{ }\mu\text{m}$. Figure R7 shows the zoom-in image of the tip. In such a cell, when Li^+ ion is fully depleted at the tip (Figure R7a), the current density concentrates at the tip (Figure R7b). At this situation, the highest cation-anion concentration difference is observed at the electrode tip (Figure R7d), which is still smaller than 0.1 mmol/L and can be negligible in our experiment. Figure R7 is also presented as Supplementary Figure S2.

Figure R7. The comsol simulations. (a) The concentration map of Li^+ at depletion; (b) current density distribution; (c) Voltage distribution in the electrolytic cell. (d) The concentration difference between cation and anion around the electrode tip.

4. Simulation based on experimental data

In addition to simulation on an ideal microelectrode, we also examine our real sample (Figure 3) by phase field simulation. The electrostatic potential in electrolyte, electrical field, and concentration difference ($C_+ - C_-$) are presented in Figure R8. The concentration difference is derived from electrical field based on the Poisson equation.⁵⁻⁶:

$$\nabla^2 \Phi = -\frac{F}{\epsilon \epsilon_0} (c_+ - c_-) \quad (\text{Equation R4})$$

where F is the Faraday constant (96485 C/mol), ϵ is relative permittivity of electrolyte and ϵ_0 is vacuum permittivity ($8.85 \times 10^{-12} \text{ F}\cdot\text{m}^{-1}$), and c_+ and c_- are concentrations of all cations and anions, respectively. Clearly the strong electrical field usually appears near the electrode, especially the tips. In these three cases, the maximum values of 40, 70 and 105 min are: $1.8 \times 10^{10} \text{ V/m}^2$, $5.4 \times 10^{10} \text{ V/m}^2$, and $2.0 \times 10^{11} \text{ V/m}^2$, respectively. If a relative vacuum permittivity of 7.8 is considered, the concentration difference ($c_- - c_+$) introduced by electrical field will be $1.2 \times 10^{-5} \text{ mmol/L}$, $3.8 \times 10^{-5} \text{ mmol/L}$ and $1.4 \times 10^{-4} \text{ mmol/L}$, much lower than the resolution of SRS.

In summary, considering the resolution of SRS, the electroneutrality hypothesis stands in our manuscript from both experimental and simulation proof.

Figure R8. The phase field simulations based on the data from Figure 3 in the manuscript. From top row to bottom row, it shows information of electrostatic potential, electrostatic field and cation-anion concentration difference at 40 minutes, 70 minutes and 105 minutes.

2. Related to point 1, the paper reads as if the authors are directly measuring Li^+ concentrations (see title, y axis of Figure 1f, 2c, 2f, etc). Particularly for readers not familiar with Raman spectroscopy, this could lead to serious confusion. The authors should be straightforward and acknowledge more directly that there are no lithium ion signatures in the Raman spectrum throughout the entire manuscript.

As we think electroneutrality is valid, it is convenient for readers to keep in mind that the measured BOB^- values in paper reflect Li^+ . We change Li^+ concentration in all Figure to $[\text{Li}^+] = [\text{BOB}^-] / M$. Also to reminder readers about this, we add the following sentence on page 6 in the main text: “It should be noticed that all $[\text{Li}^+]$ in this study is measured based on $[\text{BOB}^-]$, and their difference is small enough to be neglected at given resolutions of SRS microscopy (Supplementary note 2)”.

Reference

1. Rey, I.; Bruneel, J. L.; Grondin, J.; Servant, L.; Lassègues, J. C., Raman spectroelectrochemistry of a lithium/polymer electrolyte symmetric cell. *Journal of the Electrochemical Society* **1998**, *145* (9), 3034-3042.
2. He, R.; Xu, Y.; Zhang, L.; Ma, S.; Wang, X.; Ye, D.; Ji, M., Dual-phase stimulated Raman scattering microscopy for real-time two-color imaging. *Optica* **2017**, *4* (1), 44-47.
3. Smith, C. P., Theory of the voltammetric response of electrodes of submicron dimensions. Violation of electroneutrality in the presence of excess supporting electrolyte. *Analytical chemistry (Washington)* **1993**, *65* (23), 3343-3353.
4. Wohlfarth, C., Static dielectric constant of triethylene glycol dimethyl ether. In *Static Dielectric Constants of Pure Liquids and Binary Liquid Mixtures*, Springer: 2015; pp 185-185.
5. Feldberg, S. W., On the dilemma of the use of the electroneutrality constraint in electrochemical calculations. *Electrochemistry Communications* **2000**, *2* (7), 453-456.
6. Newman, J.; Thomas-Alyea, K. E., *Electrochemical systems*. John Wiley & Sons: 2012.

Reviewers' comments:

Reviewer #1 (Remarks to the Author):

I am satisfied with the authors' responses.

Reviewer #2 (Remarks to the Author):

In response to the unanimous and significant concern by the referees that monitoring the counterion concentration is not equivalent to monitoring lithium ion concentration, the authors have merely added two sentences to their manuscript, one of which only lists information found in the SI. In my opinion this does not adequately address this very serious concern. This is significant, as readers who are not experts in Raman spectroscopy will then be led to the erroneous conclusion that it is possible to measure monoatomic ion concentrations with a vibrational spectroscopy. As such, the paper is only suitable for a journal with a readership of experts in vibrational spectroscopy, and does not include the broad readership of Nature Communications.

Since the authors did not seriously address these concerns in this revision, only making changes to the SI, I will list several specific examples of how I believe the manuscript needs to be changed.

The title claims "visualization of ion depletion". This is incorrect – they are actually visualizing counterion growth. The concept of an "indirect measurement" needs to be included in the title.

The first sentence of the abstract claims "direct visualization of ion transport....". However this paper is all about lithium ion transport, and by no means could these SRS measurements be considered direct visualization. This argument needs to be removed in the abstract and throughout the paper.

The abstract needs a sentence stated that monitoring of lithium ions is not possible, and here the authors make the major assumption that the counterion concentration is equivalent to the lithium ion concentration, which likely does not hold under many active battery conditions, and is only accurate to within 1 mM.

The introduction needs to more thoroughly detail the limitations of the assumption, as well as the limitations of accurately determining electrolyte concentrations.

The word "directly" must be removed from the first sentence of the concluding paragraph, and at all other instances of this manuscript. Under no circumstances is this a direct measurement of lithium ion concentration.

As such, this paper has not been revised in response to previous comments, and I do not recommend for publication.

Response to the Reviewer (NCOMMS-17-32795A)

Title: “*Operando*, Three Dimensional, and Simultaneous Visualization of Ion Depletion and Lithium Growth by Stimulated Raman Scattering Microscopy”

Dear Reviewer:

First of all, we really appreciate Reviewer 2’s efforts on reviewing our manuscript and we take the reviewer’s precious comments very, very seriously. It is our negligence that we didn’t exhibit such critical points clearly in the manuscript. Thanks to the reviewer, we can further polish and improve our manuscript.

The reviewer has two major points in the second round of comments: 1) Li^+ cannot be directly measured by Raman. 2) Electroneutrality is not valid in real battery electrodes. The short answer is: 1) Li^+ can be directly measured using Raman by Li-solvent interaction, which is also used in previous literature.¹⁻⁴ 2) The applicable range of electroneutrality is explicitly discussed in the main text, and we conclude that electroneutrality can be considered valid in real battery electrodes for resolution discussed in the paper (1 mM, 100 nm). We will discuss the details below.

I. In response to the unanimous and significant concern by the referees that monitoring the counterion concentration is not equivalent to monitoring lithium ion concentration, the authors have merely added two sentences to their manuscript, one of which only lists information found in the SI.

Answer: We apologize for not explaining electroneutrality comprehensively enough in the main text, due to our immature thought of putting all discussions about electroneutrality in supporting information (2000 words and 8 pages, SI Page 4-8, 13-15). Our original thought is simply not to make the paper too long, instead of neglecting the reviewer's concerns. To correct this mistake, now a new figure (Figure 2), a new paragraph and several discussions (page 5-6 in the main text) are added into the main text related to this point. **These figures and discussions include efforts on experiments, simulations and analytical solutions on micro/nano-electrodes.**

The detailed discussions on these studies are shown in **Appendix on page 6** of this response, as they are very long. Here is a summary of our results:

1. Experimentally the direct comparison of $[Li^+]$ by Li^+ -solvent interaction (725 cm^{-1}) and $[BOB^-]$ by $C=O$ (1830 cm^{-1}) in Spontaneous Raman show that they are the same even near microelectrodes. **The difference is well below noise level ($\sim 10\text{ mM}$).**

2. Analytical solution shows that even if the electrode is 10 nm, and we measure 100 nm away from the electrode/electrolyte interface, the difference between $[Li^+]$ and $[BOB^-]$ is **smaller than $2 * 10^{-3}\text{ mM}$.**

3. COMSOL Simulation shows that the difference between $[Li^+]$ and $[BOB^-]$ near a 10 nm nanoelectrode tip is **less than 0.1 mM.**

4. Phase field simulation on our experimental results shows that the maximum difference near electrode surface is **$< 2 * 10^{-4}\text{ mM}$.**

So we can conclude that $[Li^+]$ can be considered equal to $[BOB^-]$ at given resolution SRS (500 nm and 10 mM), as the wildest result above will only introduce 1% error (0.1 mM/10 mM). To clarify this and make sure readers know such differences, following words have been added to abstract, manuscript, captions and conclusion to caution readers on this:

Abstract: *“In Raman, $[Li^+]$ can be probed by either Li^+ -solvent interaction or vibrational mode in the anion. The anion approach is used in this report. Based on "electroneutrality", the concentration difference between Li^+ and anion should not deviate for more than 0.1 mM, even near nano-electrodes.”*

Page 5: *“Therefore, in this study, $[Li^+]$ in the electrolyte is reflected by local Raman intensity of BOB^- , since at a scale of 100 nm or above, the difference between $[Li^+]$ and $[BOB^-]$ is much less than 0.1 mM, which is much smaller than our resolution ($\sim 10\text{ mM}$).”*

Page 6: *“Even on nanoelectrodes with diameter of 10 nm, both analytical solution and simulations show that the concentration difference between Li^+ and BOB^- is $< 0.1\text{ mM}$ (Supplementary Note 2, Figure 2b).”*

“For simplicity, we will use $[Li^+]$ instead of $[BOB^-]$ in the rest of the text, as evidence above indicates that such replacement will not cause any distortion of our results, but it should be noted that there may be a difference of up to 0.1 mM between them.”

Page 7: “It should be noticed that all $[Li^+]$ in this study is measured based on $[BOB^-]$, and their difference should be less than 0.1 mM, much smaller than resolution of SRS in the electrolyte (Supplementary note 2).”

Captions of Figure 3 and Figure 4: “It should be noticed that all $[Li^+]$ in this study is measured based on $[BOB^-]$, and their difference should be less than 0.1 mM, much smaller than resolution of SRS in the electrolyte.”

Figure 3, Figure 4 and Figure 5: The y-axis label changes from Li^+ concentration to $[Li^+] = [BOB^-]$

Conclusion at Page 16: “with a resolution of 10 mM, 500 nm, and $< 1s/frame$, since “electroneutrality” requires their concentrations differentiates within 0.1 mM at the spatial scale above.”

2. In my opinion this does not adequately address this very serious concern. This is significant, as readers who are not experts in Raman spectroscopy will then be led to the erroneous conclusion that it is possible to measure monoatomic ion concentrations with a vibrational spectroscopy. As such, the paper is only suitable for a journal with a readership of experts in vibrational spectroscopy, and does not include the broad readership of Nature Communications.

Answer: It is actually feasible and accurate to measure monoatomic ion concentration by ion-solvent interaction with a vibrational spectroscopy (e.g. Li^+ -EC¹ at 725 cm^{-1} , the one shown in the revised manuscript, Li^+ -DMC² at 530 cm^{-1} , Li^+ -PEO³ interaction at 745 cm^{-1} , Na^+ -DMF⁴ at 665 cm^{-1}). Li^+ cannot independently exist in the solution; it has to be associated with solvent molecules, and thus such bonds can be easily captured by Raman Spectroscopy to reflect local lithium ion concentrations. The intensity of such ion-solvent peak is proportional to cation ion concentration and this strategy has also been used widely in literature.¹⁻⁴

The only reason why we have to choose BOB^- (1830 cm^{-1}) over Li^+ -EC (725 cm^{-1}) is that the current SRS setting at Columbia only allow a **detection window** ranging from **1000 to 3300 cm^{-1}** , due to the tunable range of laser wavelength, but not the intrinsic capability of Stimulated Raman detection. To realize the visualization of Li^+ ion concentration, we can simply use a new optical parametric oscillator (OPO) to extend detection window (but it costs \$200,000). To fit the broad readership of Nature Communications, we add following words on page 5 in manuscript to help readers understand better:

“Although $[Li^+]$ can be quantitatively detected by Raman peaks of Li^+ - solvent interactions (e.g. Li^+ -ethylene carbonate at 725 cm^{-1}), these peaks are typically below 1000 cm^{-1} , which is out of the detection

¹ Journal of Power Sources 359 (2017): 435-440.

² The journal of physical chemistry letters 5.11 (2014): 2007-2011.

³ Journal of the Electrochemical Society 145.9 (1998): 3034-3042.

⁴ Journal of Raman Spectroscopy 34.6 (2003): 465-470.

window of existing setup in authors' lab. Therefore, in this study, $[Li^+]$ in the electrolyte is reflected by local Raman intensity of C=O bond in BOB⁻ at 1830 cm⁻¹, since at a scale of 100 nm or above, the difference between $[Li^+]$ and $[BOB^-]$ is smaller than 0.1 mM, much less than SRS resolution.”

3. Since the authors did not seriously address these concerns in this revision, only making changes to the SI, I will list several specific examples of how I believe the manuscript needs to be changed. The title claims “visualization of ion depletion”. This is incorrect – they are actually **visualizing counterion growth**. The concept of an “indirect measurement” needs to be included in the title.

Answer: The counterion concentration actually decreases and depletes near Li electrode, not increases, then we deduce that the concentration of Li⁺ also decrease and deplete using electroneutrality ($[BOB^-] = [Li^+]$). This phenomenon is also verified by our experiments by spontaneous Raman, which can track $[BOB^-]$ at 1830 cm⁻¹ and $[Li^+]$ at 725 cm⁻¹ simultaneously, as shown in Figure R1 below. It is also added to **Figure 2 in page 5** of main text along with other results to prove electroneutrality. More details can be found at **the Appendix** at the end of response (Figure R4, Page 9).

Figure R1. Simultaneous tracking of Li⁺ solvation peak (725 cm⁻¹) and BOB⁻ peak (1830 cm⁻¹) near electrode during lithium electrodeposition. (a) Raman Spectra taken from the start of deposition (0 min) to the end of the deposition (65 min). The intensity of both peak decreases over time. (b) The concentration changes of Li⁺ and BOB⁻ versus time.

To take reviewer 2's opinion into account, we change the title from “visualization of ion depletion” to “**visualization of anion depletion**”. We will accept reviewer's further suggestions if there are still concerns on accuracy of the title.

4. The first sentence of the abstract claims “direct visualization of ion transport...”. However, this paper is all about lithium ion transport, and by no means could these SRS measurements be considered direct visualization. This argument needs to be removed in the abstract and throughout the paper.

Answer: We have removed the word “direct” from the abstract and other places in the manuscript according to the reviewer’s suggestion. However, as stated in the answer to point 2, Li^+ can be directly seen by Li^+ -solvent interactions. It is just outside the range of the existing SRS microscopy in authors’ lab.

5. The abstract needs a sentence stated that monitoring of lithium ions is not possible, and here the authors make the major assumption that the counterion concentration is equivalent to the lithium ion concentration, which likely does not hold under many active battery conditions, and is only accurate to within 1 mM.

Answer: In the answer to point 2 we have explained why monitoring of lithium ion is feasible by Li^+ -solvent interaction. Also we can approve that the concentration difference between the cations and anions is lower than 0.1 mM even under the most severe environment (See **Appendix** Figure R5, Page 10-11). As the Reviewer required, we have added *“In Raman, $[\text{Li}^+]$ can be probed by either Li^+ -solvent interaction or vibrational mode in the anion. The anion approach is used in this report. Based on “electroneutrality”, the concentration difference between Li^+ and anion should not deviate for more than 0.1 mM, even near nano-electrodes”* in abstract of the manuscript to help reader better understand.

6. The introduction needs to more thoroughly detail the limitations of the assumption, as well as the limitations of accurately determining electrolyte concentrations.

Answer: We have made specific description and limitation on electroneutrality in the manuscript.

Page 5: “Therefore, in this study, $[\text{Li}^+]$ in the electrolyte is reflected by local Raman intensity of BOB^- , since at a scale of 100 nm or above, the difference between $[\text{Li}^+]$ and $[\text{BOB}^-]$ is much less than 0.1 mM, and much less than our resolution (~10 mM).”

Page 6: “Even on nanoelectrodes with diameter of 10 nm, both analytical solution and simulations show that the concentration difference between Li^+ and BOB^- is < 0.1 mM (Supplementary Note 2, Figure 2b).”

Page 7: “It should be noticed that all $[Li^+]$ in this study is measured based on $[BOB^-]$, and their difference should be less than **0.1 mM**, much smaller than resolution of SRS in the electrolyte (Supplementary note 2).”

Conclusion at Page 16: “with a resolution of 10 mM, 500 nm, and $< 1s/frame$, since “electroneutrality” requires their concentrations differentiates within **0.1 mM** at the spatial scale above.”

7. The word "directly" must be removed from the first sentence of the concluding paragraph, and at all other instances of this manuscript. Under no circumstances is this a direct measurement of lithium ion concentration.

Answer: We agree with the reviewer and remove all the word “directly” or “direct” in discussing visualization or measurement of Li^+ from the manuscript, such as

Abstract: “~~Direct~~ Visualization of ion transport in electrolyte”

“Utilizing this technique, we examined and provided ~~direct~~ evidence for a long-lasting question”

Page 2: “However, this model has not been validated by ~~direct~~ mapping of ion concentration profile for OD electrodes yet”

Page 16: “In conclusion, the dynamic depletion of Li^+ ions on Li electrode surface is ~~directly~~-imaged by **anion concentration in SRS microscopy**”

When we mention measurement of BOB^- , or anion, direct is still maintained.

Again, we sincerely appreciate Reviewer 2’ efforts which greatly improve the quality of our manuscript, and we always take every comments from reviewer very seriously. Please do not hesitate to contact us if there are more questions. We will seriously address every concern if there is more.

Appendix- Verification of Electroneutrality

Before discussing these supportive arguments, we would like to first clarify the meaning of electroneutrality in electrochemistry and this paper. We don’t claim that the difference between $[BOB^-]$ and $[Li^+]$ is absolutely zero, but **the difference is very small, and it can be neglected at given resolutions of SRS microscopy**. As SRS has resolution of 1-10 mM and ~500 nm in our experiment, and our measurement is at least 1 μm away from electrolyte/electrode interface, the electroneutrality means

that at scale of ~ 100 nm, the difference between $[\text{BOB}^-]$ and $[\text{Li}^+]$ is much less than 1 mM. Therefore, such difference can be neglected in our analysis. This is the foundation of following discussions. The following verification includes four aspects: 1) Experimental measurement by spontaneous Raman, 2) Analytical solutions, 3) simulations on 10 nm-nanoelectrodes, and 4) phase field simulations with experimental data.

1. Experimental Proof

We don't find any established experimental method in literature to justify electroneutrality. Therefore, we develop an approach that is most solid based on our experience and discussion with electrochemists. We hope that the results can satisfy the reviewer, and we appreciate if the reviewer can provide suggestions on experiments if more validations are needed.

As the SRS setup at Columbia can only detect wavenumber higher than 1000 cm^{-1} , we cannot directly see Li^+ ion based on Li-electrolyte interaction. Therefore, we track both Li^+ and BOB^- near electrode surface by spontaneous Raman to derive their local concentration. The electrolyte used in this experiment is LiBOB / (TEGDME:EC v/ 7:3) instead of LiBOB/TEGDME, so that $[\text{Li}^+]$ can be tracked by the Li^+ -EC interaction at 725 cm^{-1} , and BOB at 1830 cm^{-1} (Figure R2a).

As shown in Figure R2b, the intensities at the designated wavenumbers have very good linearity with concentration, so that the derived $[\text{Li}^+]$ and $[\text{BOB}^-]$ have high accuracy. The standard deviation is determined to be 8.3 mM for Li^+ -EC solvation and 5.8 mM for BOB^- for a 14 second accumulation per point. It should be noted that the accumulation time is 10^6 longer than SRS to get similar chemical resolution.

Figure R2. (a) The Raman spectrum of LiBOB / (TEGDME: EC v/ 7:3) electrolyte with a concentration from 0 M to 0.4 M. The accumulation time is 14 seconds without average for each spot. (b) The plot of

counts at designated wave number (725 cm^{-1} for Lithium solvation and 1830 cm^{-1} for BOB^-) versus concentration of LiBOB.

Then we built the same lithium-lithium electrolytic cell as Figure 1c in the main text to probe local ion concentration under spontaneous Raman (XploRA One by HORIBA). Then this electrolytic cell was tested under variant current densities (from the 0.5 mA/cm^2 to 1.5 mA/cm^2 , Figure R3A) until ions at electrode surface are fully depleted together with lithium dendrite growth. The changing current density can also help verify the electroneutrality under different electrical field.

During the cell operation, lithium is deposited as dendrite and gradually approach the Raman spot which has a size $\sim 10\text{ }\mu\text{m}$. We find that when the laser directly shines at lithium, severe signal loss and distortion will occur, which may be due to interaction with SEI and lithium. Therefore, we only approach to $\sim 10\text{ }\mu\text{m}$ away from the lithium surface, which is similar with our paper (Figure R3B).

Figure R3. (a) The voltage and current curve versus time plot in a lithium- lithium electrolytic cell. (b) The optical image of electrode at the beginning and at the end of lithium electrodeposition. After the deposition, there are large amount of dendrites formed on the electrode surface and the colored square in the optical images show the position where Raman spectra were taken.

After converting Raman intensity to ion concentration, we can clearly see that $[\text{Li}^+]$ and $[\text{BOB}^-]$ are reduced due to ion depletion (Figure R4A&B). Their absolute concentrations synchronize with each other, so that they appear to be the same as each other and electroneutrality is valid.

Figure R4. (a) Raman Spectra taken from the start of deposition (0 min) to the end if the deposition (65 min). The intensity of both peak decreases over time. (b) The concentration changes of Li^+ and BOB^- versus time.

To further answer this question more quantitatively, we performed the hypothesis testing in statistics. Data at 50 minutes were used as an example here. For cations, $X_C \sim N(\mu_C, \sigma_C^2)$, X_C is the measured cation concentration 0.01532 M and σ_C is the standard deviation of cation concentration 0.00828 M. The same go with anions, $X_A \sim N(\mu_A, \sigma_A^2)$ while $X_A = 0.01455$ M and $\sigma_A = 0.0058$ M. We want to test whether the mean concentration for cations μ_C and anions μ_A are consistent. The null hypothesis is $\mu_C = \mu_A$ with a significance level of 1%.

$X_C - X_A$ is normal distributed with a mean of $\mu_C - \mu_A$ and standard deviation of $\sqrt{\sigma_C^2 + \sigma_A^2}$. For hypothesis $\mu_C - \mu_A = 0$,

$$\frac{X_C - X_A - 0}{\sqrt{\sigma_C^2 + \sigma_A^2}} \sim N(0,1) \quad (\text{Equation R1})$$

The test statistic is 0.074, far smaller than $Z_{0.005}$ ($Z_{0.005} = 2.58$, the value can be found in Z table), so the null hypothesis is accepted. Other experimental results (50 min - 65 min) with test statistics (-0.363, 0.082, -0.216, 0.148) all fall in the range from $-Z_{0.005}$ to $Z_{0.005}$.

It is worth noticing that the sum of the square of each time also follows chi-square distribution. The chi-square test statistics is $\sum \left(\frac{X_C - X_A}{\sqrt{\sigma_C^2 + \sigma_A^2}} \right)^2 = 0.182$, which is much smaller than the chi-square table at

degree of freedom of 5 at significance level of 5% (16.75). We can conclude that the concentration of cation and anion are equal in the whole experiment.

2. Analytical Solutions

The effectiveness of electroneutrality on microelectrodes was actually analyzed by Prof. Henry White at University of Utah¹. Although the title is “violation of electroneutrality”, it actually shows that the difference between cation and anion concentration ($\ll 1$ mM) is much less than our resolution, and **his “violation” simply means that when electrode is nano-sized, the concentration difference is μ M-level, which is much larger than that of flat surface (e.g. nM or below)**. In equation 9 of reference ¹, Henry White showed that under voltammetric response and outside the Debye length, the local net charge in electrolyte (Σ_{error}) near a spherical microelectrode satisfies

$$-r_0^2 \frac{\epsilon \epsilon_0}{F} \nabla^2 \phi = 18.4 \text{ nm}^2 \text{ mM} \left[\frac{r}{r_0} + \frac{2\gamma(r/r_0)^2}{I/I_l} \right]^{-2} \equiv r_0^2 (c_+ - c_-) \quad (\text{Equation R2})$$

Where r_0 is the radius of spherical microelectrode, F is the Faraday constant (96485 C/mol), ϵ is relative permittivity ($\epsilon=7.8$ for TEGDME)² and ϵ_0 is vacuum permittivity ($8.85 \cdot 10^{-12}$ F*m⁻¹), Φ is the potential difference between the position r in the solution and the bulk of the solution, γ represents the ratio of supporting electrolyte to reactant concentration, I/I_l is current after being normalized by limiting current and, c_+ / c_- are concentrations of all cations and anions respectively.

Therefore, the maximum charge difference occurs at $\gamma = 0$, which means no supporting electrolyte to shield electrical field. This is also the same as our experimental condition, where the electrolyte is binary without supporting electrolyte ($\gamma = 0$). In this case, the second term in the square brackets is zero and the equation can be simplified to

$$18.4(\text{nm}^2 \text{ mM})r^{-2} \equiv (c_+ - c_-) \quad (\text{Equation R3})$$

Based on equation 3, the difference between $[\text{Li}^+]$ and $[\text{BOB}^-]$ is purely determined by the distance to the center of the microelectrode. Therefore, r could be much smaller in microelectrode than bulk electrode. **Even if the electrode size is 10 nm, and we measure 100 nm away from the electrode/electrolyte interface, $c_+ - c_- = 18.4 / (110^2) = 1.52 \cdot 10^{-3}$ mM, which is much less than SRS resolution.** In addition, the typical Debye length is ~ 5 nm at the concentration of 1 mM, so 100 nm satisfy the prerequisite of outside Debye region.

3. Simulation on microelectrode.

To further validate the analytical solution that C_+-C_- is less than 1 mM. COMSOL simulation on microelectrode is performed. The parameters used are: Diffusion coefficient is $5 \cdot 10^{-7} \text{ cm}^2/\text{s}$, current density is $2 \text{ mA}/\text{cm}^2$, and electrode distance is $100 \text{ }\mu\text{m}$. The electrode is designed to have a single tip with **a width of 10 nm** and a length of $2 \text{ }\mu\text{m}$. Figure R5 shows the zoom-in image of the tip. In such a cell, when Li^+ ion is fully depleted at the tip (Figure R5a), the current density are mainly near the tips (Figure R5b) and the voltage decay more near the tip (Figure R5c). At this situation, the highest cation-anion concentration difference is observed at the electrode tip), which is still smaller than $0.1 \text{ mmol}/\text{L}$ and can be negligible in our experiment.

Figure R5. The comsol simulations. (a) the concentration map of Li^+ at depletion; (b) current density distribution; (c) Voltage distribution in the electrolytic cell. (d) the concentration difference between cation and anion around the electrode tip.

4. Simulation based on experimental data

In addition to simulation on an ideal microelectrode, we also examine our real sample (Figure 4) by phase field simulation. The electrostatic potential in electrolyte, electrical field, and concentration difference ($C_+ - C_-$) are presented in Figure R6. The concentration difference is derived from electrical field based on the Poisson equation.³⁻⁴:

$$\nabla^2 \Phi = -\frac{F}{\epsilon \epsilon_0} (c_+ - c_-) \quad (\text{Equation R4})$$

where F is the Faraday constant (96485 C/mol), ϵ is relative permittivity of electrolyte and ϵ_0 is vacuum permittivity ($8.85 \times 10^{-12} \text{ F} \cdot \text{m}^{-1}$), and c_+ and c_- are concentrations of all cations and anions, respectively. Clearly the strong electrical field usually appears near the electrode, especially the tips. In these three cases, the maximum values of 40, 70 and 105 min are: $1.8 \times 10^{10} \text{ V/m}^2$, $5.4 \times 10^{10} \text{ V/m}^2$, and $2.0 \times 10^{11} \text{ V/m}^2$, respectively. If a relative vacuum permittivity of 7.8 is considered, the concentration difference ($c_+ - c_-$) introduced by electrical field will be $1.2 \times 10^{-5} \text{ mmol/L}$, $3.8 \times 10^{-5} \text{ mmol/L}$ and $1.4 \times 10^{-4} \text{ mmol/L}$, much lower than the resolution of SRS.

In summary, considering the resolution of SRS, the electroneutrality hypothesis should stand in our manuscript from the experimental and simulated proof.

Figure R6. The phase field simulations based on the data from Figure 3 in the manuscript. From top row to bottom row, it shows information of electrostatic potential, electrostatic field and laplacian of cation-anion concentration difference at 40 minutes, 70 minutes and 105 minutes.

1. Smith, C. P., Theory of the voltammetric response of electrodes of submicron dimensions. Violation of electroneutrality in the presence of excess supporting electrolyte. *Analytical chemistry (Washington)* **1993**, *65* (23), 3343-3353.
2. Wohlfarth, C., Static dielectric constant of triethylene glycol dimethyl ether. In *Static Dielectric Constants of Pure Liquids and Binary Liquid Mixtures*, Springer: 2015; pp 185-185.
3. Feldberg, S. W., On the dilemma of the use of the electroneutrality constraint in electrochemical calculations. *Electrochemistry Communications* **2000**, *2* (7), 453-456.
4. Newman, J.; Thomas-Alyea, K. E., *Electrochemical systems*. John Wiley & Sons: 2012.

REVIEWERS' COMMENTS:

Reviewer #3 (Remarks to the Author):

I reviewed the revised manuscript and also read the answers to the reviewers. The work is very interesting and novel for the field of batteries. The method suggested by the authors can give many new information on the lithium growth and can be extended to many systems. There are still many issues of language in spite of the editing already done. The structure of the sentences is too complicated, sentences are too long and too many repetitions. Somehow, the manuscript has to be simplified and polished. There are typos and errors which have to be corrected.

Reviewer #4 (Remarks to the Author):

The author demonstrate that SRS is a powerful tool to monitor the concentration of anion in an electrolyte, which is equal to Li^+ concentration within the accuracy required in the present work. High temporal and spatial resolutions and chemical sensitivity were achieved in analysis of electrolyte near a Li electrode during electrodeposition of Li on it. They obtained 3-dimensional mapping of the concentration that changes during Li deposition. They revealed detailed correlation between ionic concentration and growth rates that is difficult to observe by other methods, that is, emergence of positive feedback mechanisms between them. Based on the results, they also demonstrated two approaches aiming uniform Li deposition which is required for operation of various batteries. I feel the paper will influence thinking in development of homogeneous Li deposition processes in the battery community. I confirmed the validity of the statistical analysis and the experimental procedures.

In summary, I feel that the quality, novelty and interest to researches in other fields of the revised paper are sufficient. I recommend the paper for publication in Nature Communications.

Reviewer's comments:

Reviewer #3 (Remarks to the Author):

I reviewed the revised manuscript and also read the answers to the reviewers. The work is very interesting and novel for the field of batteries. The method suggested by the authors can give many new information on the lithium growth and can be extended to many systems. There are still many issues of language in spite of the editing already done.

Answer: We greatly thank the reviewer 3's positive comments and precious advice. To address reviewer 3's concern on languages, we have thoroughly revised our manuscript with the help of Nature English language editing: gold service and colleagues who are native English speakers. We have further revised our manuscript based on the manuscript checklist for more natural understanding.

The structure of the sentences is too complicated, sentences are too long and too many repetitions. Somehow, the manuscript has to be simplified and polished. There are typos and errors which have to be corrected.

Answer: We have shortened the sentence as much as possible; meanwhile, we have cut ~ 500 words to simplify our manuscript without changing the essential information. For example:

At Page 6: Before revision: “This indicates that SRS microscopy is sensitive enough to resolve the minute concentration change during Li deposition, thus laying a solid foundation for our imaging and quantitative analysis below. In addition to the high sensitivity, the fast acquisition speed provided by SRS allows one 3D volumetric image to be taken within 10 seconds.”

After revision: “These results indicate the high sensitivity of SRS microscopy and lay a solid foundation for our imaging and quantitative analysis below. Such high sensitivity allows a 3D volumetric image to be taken within 10 s.”

At Page 6: after the sentence: “We subsequently confirmed that SRS measurements of LiBOB depletion near a Li metal surface are barely affected by presence of Li dendrites (Supplementary Fig. 7).” We delete “~~These desired features render our method a dual-modality imaging technique to simultaneously visualize both liquid and solid phases all at once.~~” to reduce repetition.

At Page 7: the concentration overpotential is low, and the corresponding voltage profile is nearly flat ~~with a slight increase in this stage.~~

Reviewer #4 (Remarks to the Author):

The author demonstrate that SRS is a powerful tool to monitor the concentration of anion in an electrolyte, which is equal to Li⁺ concentration within the accuracy required in the present work. High temporal and spatial resolutions and chemical sensitivity were achieved in analysis of electrolyte near a Li electrode during electrodeposition of Li on it. They obtained 3-dimensional mapping of the concentration that changes during Li deposition. They revealed detailed correlation between ionic concentration and growth rates that is difficult to observe by other methods, that is, emergence of positive feedback mechanisms between them. Based on the results, they also demonstrated two approaches aiming uniform Li deposition which is required for operation of various batteries. I feel the paper will influence thinking in development of homogeneous Li deposition processes in the battery community. I confirmed the validity of the statistical analysis and the experimental procedures. In summary, I feel that the quality, novelty and interest to researches in other fields of the revised paper are sufficient. I recommend the paper for publication in Nature Communications.

Answer: We are delighted that the reviewer is satisfied with our work. We want to thank the reviewer for your time on reviewing our paper and greatly appreciate Reviewer 4's positive feedback.

Again, we greatly appreciate your time on reviewing our manuscript and constructive comments. Please do not hesitate to contact us if you have any other concerns.